# *Perception Test*: A Diagnostic Benchmark for Multimodal Video Models

**Viorica Pătrăucean**[1][*]          **Lucas Smaira**          **Ankush Gupta**          **Adrià Recasens Continente**
DeepMind                    DeepMind                    DeepMind                    DeepMind

**Larisa Markeeva**          **Dylan Banarse**          **Skanda Koppula**          **Joseph Heyward**
DeepMind                    DeepMind                    DeepMind                    DeepMind

**Mateusz Malinowski**          **Yi Yang**          **Carl Doersch**          **Tatiana Matejovicova**          **Yury Sulsky**
DeepMind                    DeepMind          DeepMind          DeepMind                    DeepMind

**Antoine Miech**          **Alex Frechette**          **Hanna Klimczak**          **Raphael Koster**          **Junlin Zhang**
DeepMind                    DeepMind                    DeepMind                    DeepMind                    DeepMind

**Stephanie Winkler**          **Yusuf Aytar**          **Simon Osindero**          **Dima Damen**
DeepMind                    DeepMind                    DeepMind                    University of Bristol

**Andrew Zisserman**                    **João Carreira**[1]
University of Oxford, DeepMind                    DeepMind

## Abstract

We propose a novel multimodal video benchmark – the *Perception Test* – to evaluate the perception and reasoning skills of pre-trained multimodal models (e.g. Flamingo, SeViLA, or GPT-4). Compared to existing benchmarks that focus on *computational tasks* (e.g. classification, detection or tracking), the *Perception Test* focuses on *skills* (Memory, Abstraction, Physics, Semantics) and *types of reasoning* (descriptive, explanatory, predictive, counterfactual) across video, audio, and text modalities, to provide a comprehensive and efficient evaluation tool. The benchmark probes pre-trained models for their *transfer* capabilities, in a zero-shot / few-shot or limited finetuning regime. For these purposes, the *Perception Test* introduces 11.6k real-world videos, 23s average length, designed to show perceptually interesting situations, filmed by around 100 participants worldwide. The videos are densely annotated with six types of labels (multiple-choice and grounded video question-answers, object and point tracks, temporal action and sound segments), enabling both language and non-language evaluations. The fine-tuning and validation splits of the benchmark are publicly available (CC-BY license), in addition to a challenge server with a held-out test split. Human baseline results compared to state-of-the-art video QA models show a substantial gap in performance (91.4% vs 46.2%), suggesting that there is significant room for improvement in multimodal video understanding. Dataset, baselines code, and challenge server are available at https://github.com/deepmind/perception_test

---

[*]Corresponding author: viorica@google.com, [1]shared senior contribution

37th Conference on Neural Information Processing Systems (NeurIPS 2023) Track on Datasets and Benchmarks.

# 1 Introduction

Significant progress in multimodal models has been made recently due to large-scale training on multimodal data. Models like Flamingo [4], SeViLA [55], BEiT-3 [49], GPT-4 [43] show remarkable versatility, dealing with diverse data sources and tackling new tasks by observing only a handful of examples. This is a major departure from specialised models that are typical in computer vision, *e.g.* image or action classifiers [53, 20], object detectors [13], or object trackers [47], opening up the path towards general perception and reasoning models.

Benchmarking these models in a robust and efficient way is key to expanding their capabilities, by allowing researchers to rank model design and training choices, and identify areas for improvement. Many perception-related benchmarks exist, for example Imagenet for image classification [17], Kinetics for video action recognition [36], Audioset for audio event classification [24], TAO for object tracking [16], or VQA for image question-answering [28], to name only a few. While these benchmarks have led to amazing progress, they all target restricted aspects of perception, focusing on specific computational tasks: *e.g.* image benchmarks discard the temporal dimension, visual question-answering tends to focus on only high-level semantic scene understanding, and object tracking focuses on lower-level, texture-based cues. Gluing several datasets together [39, 45] to benchmark more general models (as is done in Flamingo, SeViLA, BEiT-3, or GPT-4) improves coverage, but results in an expensive evaluation procedure that still misses important general perception abilities, *e.g.* physics understanding or memory. Few existing benchmarks even define tasks over both audio and visual modalities [29], much less more complex combinations of modalities and tasks. Furthermore, most prior work provides large training sets and thus benchmark models for in-dataset capabilities.

In this work, we propose the *Perception Test* – a benchmark formed of purposefully designed, filmed, and annotated real-world videos that aims to comprehensively assess the capabilities of multimodal perception models across different skill areas (Memory, Abstraction, Physics, Semantics), types of reasoning [54] (*descriptive*, *explanatory*, *predictive*, and *counterfactual*), and modalities (video, audio, text). Our benchmark draws inspiration from diagnostic synthetic datasets like CATER [25] or CLEVRER [54], behavioral tests like the Visual Turing Test [41, 23], experiments in developmental psychology [1, 6, 31], and motor-free perception screening tests used for children or adults [42, 22].

To avoid benchmark overfitting, we propose a generalisation-focused evaluation regime. We aim to benchmark any representation or model, pre-trained with any *external* dataset or task, of any scale available. The *Perception Test* itself contains a small training set that can optionally be used for fine-tuning task decoders or prompting the model, and the rest is used for evaluation (public validation and held out test sets). In this regime, we can more robustly assess the *transfer* abilities of these models, such that improvement on the benchmark can more reliably predict generalisation to real-world operation.

The dataset contains 11.6K real-world videos, densely annotated with 190K object and 8.6K point tracks, 73.5K temporal action segments, 137K temporal sound segments, 38K multiple-choice video question-answer (mc-vQA) pairs and 6K grounded video question-answer (g-vQA) pairs, enabling both language and non-language evaluations, to ensure a thorough assessment; see Figure 1 and Table 3. Having multiple types of annotations per video is useful also for analysis and explainability purposes, as the correlations between successes and failures across tasks may uncover model biases.

We open-source the videos and annotations in the training and validation splits. An evaluation server is made available together with the videos from the held-out test split. Since currently there is no model that can tackle all the evaluation tasks in our benchmark, we provide baseline results for per-task models: object tracking, point tracking, temporal action localisation, temporal sound localisation, multiple-choice video question-answering, and grounded video question-answering. For the mc-vQA task, the performance is mapped across skill areas (memory, abstraction, physics, semantics), and types of reasoning (descriptive, explanatory, predictive, counterfactual) to obtain a comprehensive diagnostics report.

In the next section (section 2), we discuss related work in more detail, highlighting what sets the *Perception Test* apart in the rich landscape of multimodal benchmarks. In sections 3 and 4, we describe the videos and annotations in the *Perception Test*, with details about the diversity of participants involved in filming the videos. In section 5, we introduce the computational tasks enabled by these annotations, together with evaluation metrics and baselines, including a human baseline. We conclude with a summary and directions of future work in section 6.

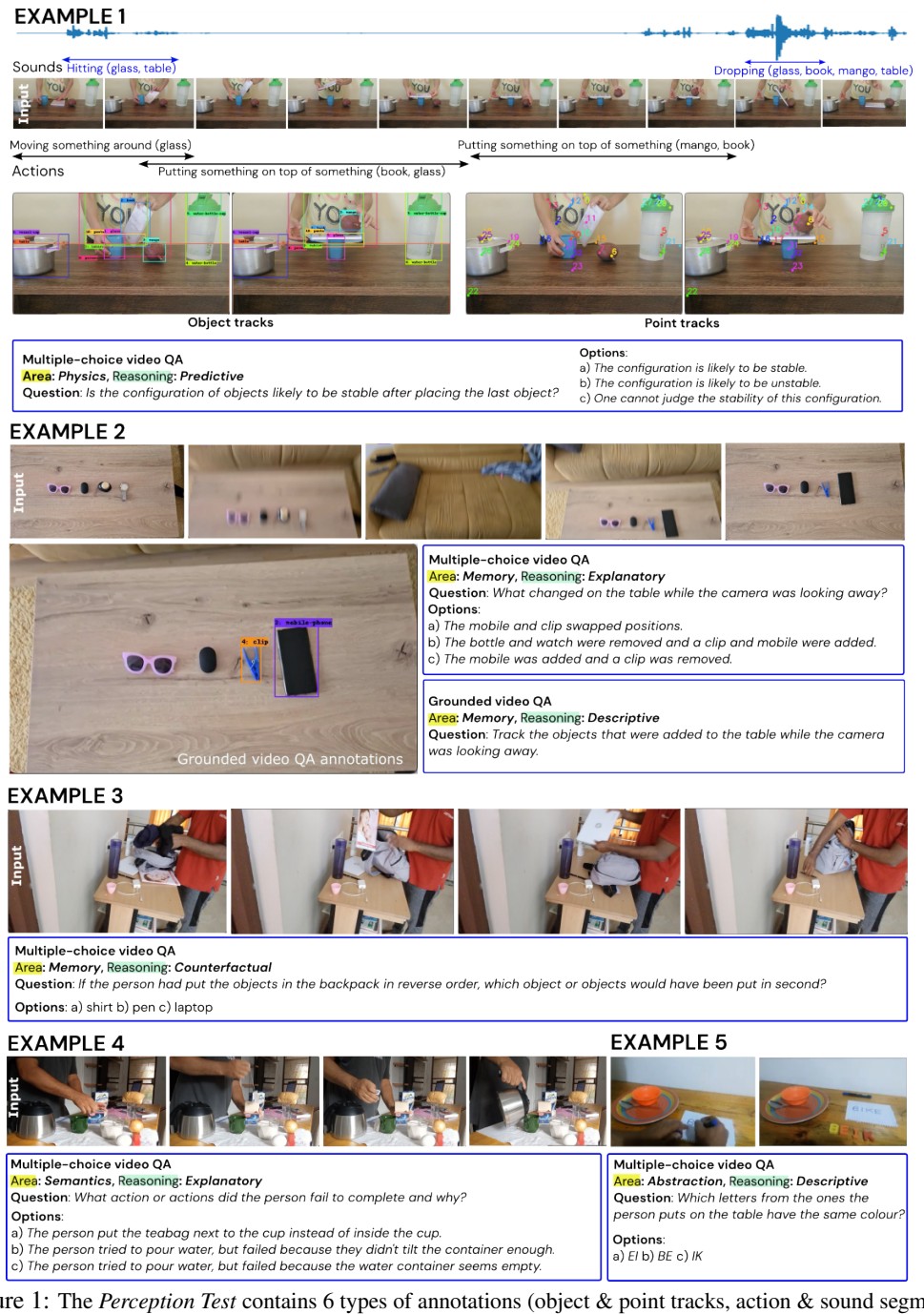

Figure 1: The *Perception Test* contains 6 types of annotations (object & point tracks, action & sound segments, multiple-choice videoQA and grounded videoQA) and tasks spanning 4 skill areas (Memory, Asbtraction, Physics, Semantics, and 4 types of reasoning (Descriptive, Explanatory, Predictive, Counterfactual). See the presentation video at `https://github.com/deepmind/perception_test` for more examples.

## 2 Related work

A large number of perception-related benchmarks exist in the literature, covering various computational tasks or modalities. We focus the discussion here on video benchmarks and highlight the differences between the *Perception Test* and prior work, in terms of data collection process, covered modalities, and available annotations and tasks.

| Dataset | Source | Skills | # videos | Dens | L(s) |
|---|---|---|---|---|---|
| Charades | C,R | S | 10,000 | 14 | 30 |
| SSv2 | C,R | AS | 108,499 | 1 | 4 |
| Ego4D-v2 | R | MS | 205,534$^{\ddagger}$ | 9$^{*}$ | 492$^{\dagger}$ |
| CLEVRER$^{\flat}$ | C,Y | P | 60,000 | N/A | 5 |
| *Perception Test* | C,R | MAPS | 11,620 | 761 | 23 |

Table 1: Characteristics of different datasets compared to the *Perception Test*. Dataset sources: Scripted (C), Real (R) and Synthetic (Y). Skill areas: Memory (M), Abstraction (A), Physics (P), Semantics (S). Dens: Average number of annotations per video. L: Average video length in seconds. $^{\ddagger}$number of annotated clips, $^{*}$reported for hand-objects subset with the highest density of annotations, $^{\dagger}$reported for ELM NLQ subset with highest average clip length. $^{\flat}$: Annotations are extracted directly from the simulator.

Existing real-world benchmarks rely on one of the following data sources: **(i)** Videos collected from the web or repositories like Youtube, *e.g.* Kinetics [36], ActivityNet [9], VGGSound [11], HVU [18], ActivityNet-QA [56], tGIFQA [33]; **(ii)** Videos collected on demand, filmed by volunteers doing arbitrary activities in indoor or outdoor scenes, *e.g.* EPIC-KITCHENS [15], Ego4D [29]; **(iii)** Videos collected on demand, filmed by crowd-sourced participants doing actions described in pre-defined scripts, mostly in indoor scenes, *e.g.* Charades [46], Something-Something v2 (SSv2) [26]. Invariably, all real-world benchmarks use crowd-sourced annotations to enable various computational tasks like action classification, object detection, or video captioning, to name only a few.

Annotating publicly available videos is useful for training. However, using this approach for general perception evaluation has multiple drawbacks. Large quantities of data would need to be amassed and carefully filtered and annotated to accumulate (statistically) sufficiently diverse samples showing perceptually interesting situations that require skills like memory, abstraction, physics, and semantics understanding. In addition, some types of data are simply not available, *e.g.* situations showing incorrect execution of simple tasks like tying shoe laces. As we aim to assess more diverse skills, we chose to design video scripts that show perceptually interesting and diverse situations and film these with crowd-sourced participants from different places in the world to ensure diversity of video content and appearance. Different from Charades where the scripts were designed by crowd-sourced workers, our scripts are designed by our research team, similar to Something-something (v2). However, we did not aim to obtain an exhaustive coverage of simple actions like in SSv2. Instead, we designed more complex scripts to probe for more advanced reasoning skills beyond action classification.

A few research works have highlighted the need for robust diagnostics benchmarks, *e.g.* CATER [25], CLEVRER [54], IntPhys [44], Physion [7]. Their authors developed synthetic datasets to evaluate in a more systematic way, across different levels of difficulty, models' abilities to reason about intuitive physics (object collisions, motion, object permanence). We share the same motivation of creating a diagnostic test, and we aim to cover aspects related to memory, abstraction, intuitive physics, and semantics, using real-world videos. To achieve this, in addition to designing the video scripts, our team also designed the questions for each script type for the high-level tasks (mc-vQA and g-vQA); the answers per video were provided by crowd-sourced annotators. Given that our videos are filmed in real world scenes using common household items, the distributions of objects, actions, and sounds in our benchmark have a significant overlap with standard computer vision datasets (*e.g.* 99.01% of the words in our benchmark also appear in VQAv2 [27]), hence the domain gap between the *Perception Test* and existing large-scale training datasets should be minimal.

Table 1 summarises the characteristics of the *Perception Test* compared to previous efforts. It can be observed that the *Perception Test* has a better coverage of skill areas and higher density of annotations[2]. Size-wise, it is comparable to Charades, but smaller than Ego4D or SSv2. We emphasise that the *Perception Test* is not designed to be a large-scale training dataset. Instead, it is an evaluation benchmark, with limited fine-tuning or prompting data, meant to assess the transfer capabilities of models.

---

[2]We count every labeled box, point, temporal segment, or question as a separate annotation

## 3   Videos in the *Perception Test*

Inspired by how human perception screening tests are carefully designed by experts in developmental psychology or medicine (*e.g.* [12]), we designed video scripts and tasks to diagnose the perception skills of our models.

**Scripts design:** Our goal was not to obtain an exhaustive coverage of activities or types of scenes. Instead, we selected four areas – Memory, Abstraction, Physics, Semantics – within which several skills should be tested (see Table 2, first column) through tasks that require different types of reasoning: descriptive, explanatory, predictive, or counterfactual [54]. The skills selection took into account blind spots of existing benchmarks, weaknesses of current models, and aspects that are important for real-world scene understanding.

We then created scripts describing simple situations or games that can be easily performed by any one person (non-professional actor) using the items available in a regular household, or items that can be easily crafted if not available (*e.g.* letters or geometric shapes crafted from paper or cardboard). Each script consists of a brief description of the scene, followed by a description of the actions to be performed, together with specification of the camera placement (static camera one viewpoint; static camera 2 viewpoints; static camera and moving camera). To enhance content diversity, each script had considerable room for variability in the number of objects to be included in the scene or types of actions to be performed, or order of actions.

We prioritised situations where we can test high-level concepts like memory through low-level tasks like object tracking and the other way around: low-level physics understanding probed through high-level tasks like question-answering. In addition, we included in each script elements that could make the situations more interesting and challenging. For example, in cooking scripts (*e.g.* making tea, making salad), we added *distractor actions* [46], i.e. actions not relevant for making tea and that have no impact on the outcome of the making tea sequence, like clapping hands, or hitting a kettle with a spoon; this allows probing for understanding of causal relations between actions. We also included *distractor objects* in the scene description, i.e. objects that are not relevant for the current script, but which are relevant for other scripts, like tomatoes present on the table during the make tea activity [48]. For all the scripts, we also asked participants to include in the scene some *adversarial configurations of objects e.g.* a shoe on the table. This allows us to probe models for understanding of spatial relations of objects when the language biases are not valid. Finally, some of the script variations include *adversarial actions* [26], i.e. incorrectly executed actions. For example, when making the tea, all the steps are done normally, but one is incorrectly executed, like pouring water from an empty kettle. In this way, we can probe for understanding of task completion, in a more complex setup than the adversarial action classification used in SSv2 dataset [26].

Table 2 and Figure 1 show examples of situations included in the scripts to probe for different skills in the different areas and types of reasoning. Note that the videos associated with a script allow defining tasks and questions across multiple skill areas. All-in-all, we designed 37 scripts, each with 2-5 variations, to obtain a diverse dataset. Having multiple variations per script allows us to ask the exact same question with the same set of options, and the correct answer depends on the specific script variation – in this way, we can avoid language biases in questions that give away the answer [37]. Examples of videos included in the dataset can be found in the presentation video at https://github.com/deepmind/perception_test.

**Video filming:** Ensuring diversity of participants and scenes depicted in the videos was a critical consideration when developing the benchmark. Using a crowdsourcing pool, we selected around 100 participants from different countries of different ethnicity and gender and aimed to have a diverse representation within each video script. We include in the appendix details about the self-reported demographics of participants. Each script variation was filmed by at least a dozen of different participants, using most often a mobile-phone camera, resulting in high-resolution audio-visual assets. For scripts to be filmed from two different viewpoints, the recording was most often done sequentially by repeating the script; a few participants recorded simultaneously using two filming devices. About 15% of the videos were filmed with a moving camera. Most of the videos were filmed indoors in the living room or kitchen, with a small number being filmed in the bathroom or outdoors (about 1%). Most of the activities are performed on a tabletop, but some are also performed on the floor or on a chair. To avoid privacy concerns, we instructed the participants to not record their faces or

| (Skill Area) Skill | Example of situations and questions or tasks |
|---|---|
| (M)Visual discrimination | Objects are shown in front of the camera, with some shown more than once. **Task**: Detect which objects were shown multiple times. |
| (M) Change detection | The camera is filming a table, then looks away for a few seconds, then looks back at the table. Some changes may have occurred. **Task**: Explain what changed. |
| (M) Sequencing | Objects are put in a backpack. **Task**: List their order. |
| (M) Event recall | A person indicates a region on the table with the hand, then puts objects inside and outside the region. **Task**: List the objects put inside the region. |
| (A) Object, action & event counting | A person turns a lamp on and off. **Task**: Count the number of times the illumination changed in the scene. |
| (A) Feature matching | A person puts wooden letters on the table. **Task**: Which letters have the same colour? |
| (A) Pattern discovery | Geometric shapes are shown in a pattern. **Task**: What shape will be shown next? |
| (A) Pattern breaking | A person puts multiple cups all facing upwards and one facing downwards. **Task**: Indicate the object that breaks the pattern. |
| (P) Object permanence | A person plays a cups-game with 3-4 cups by hiding a small object under one of the cups, then shuffles the cups. **Task**: Predict where is the hidden object after shuffling. |
| (P) Spatial relations & containment | A person puts a bookmark in a book, then puts the same or another book in a backpack. **Task**: Where is the bookmark at the end? |
| (P) Object attributes | A person writes on a piece of paper. **Task**: Is the paper lined or plain? |
| (P) Motion & occluded interactions | A person moves an occluder object in front of a small object, sometimes moving also the small (occluded) object. **Task**: Was the small object moved? |
| (P) Solidity & collisions | A person launches objects against a blocker object, sometimes removing the blocker. **Task**: Does the object fall off the table? |
| (P) Conservation | A person pours an equal amount of water in 2 identical glasses, then pours all or part of the water from one glass in a taller or wider glass. **Task**: How much water is in the last glass? |
| (P) Stability | A person puts objects on top of each other in a stable or unstable configuration. **Task**: Predict if the configuration will be stable after placing the last object. |
| (S) Distractor actions & objects | A person makes tea, and does also some distractor actions unrelated to making tea, *e.g.* rotating a knife. **Task**: Identify the distractor action(s). |
| (S) Task completion & adversarial actions | A person ties shoe laces, but sometimes pretends to tie, or ties the lace of one shoe to the lace of the other shoe. **Task**: Detect if the action is done correctly. |
| (S) Object & part recognition | A person conceals a small object in one of their hands, then shuffles the hands. **Task**: Identify in which hand is the object held. |
| (S) Action & sound recognition | All scripts. **Task**: Detect the actions and sounds in the video from a pre-defined list. |
| (S) Place recognition | All scripts. **Task**: Detect where is the action taking place. |
| (S) State recognition | A person uses an electric device. **Task**: Indicate if the device is on. |
| (S) General knowledge & Language | Some objects are shown to the camera, some multiple times. **Task**: Given a list of arbitrary statements or word puzzles, some requiring general knowledge to solve, select the statement that contains a reference to the second distinct object shown. |

Table 2: Examples of scripts probing for different skills in the four areas in the *Perception Test*: **(M)**:Memory, **(A)**:Abstraction, **(P)**:Physics, **(S)**:Semantics.

voices. This is not a limitation of the dataset since the focus in our scripts is on object interactions. The participants gave their consent for the data to be used, published, and stored for perpetuity.

**Splits:** The *Perception Test* contains 11609 videos (with audio), 23s average length. It is divided into a small training split (2184 videos, $\sim 20\%$ of the data) that can be used for fine-tuning or prompting, a validation split (5900 videos, $\sim 50\%$ of the data), and a held-out test split (3525 videos, $\sim 30\%$ of the data) available through the evaluation server. We optimised to obtain a good balance across all annotation types and camera motions across the 3 splits; see section 5 in the appendix.

## 4  Annotations in the *Perception Test*

We annotate these videos with six types of annotations to cover low-level and high-level aspects, both spatial and temporal, and enable language and non-language evaluations: object and point tracks, temporal action and sound segments, multiple-choice and grounded video question-answers. We include a summary of the number of annotations in Table 3 and visualisations in Figure 1.

| Annotation type | # classes | # annot | # videos | Rate (fps) |
|---|---|---|---|---|
| Objects tracks | 5101 | 189940 | 11609 | 1 |
| Point tracks | NA | 8647 | 145 | 30 |
| Action segments | 63 | 73503 | 11353 | 30 |
| Sound segments | 16 | 137128 | 11433 | 30 |
| mc-vQA | 132 | 38060 | 10361 | NA |
| g-vQA | 34 | 6086 | 3063 | 1 |

| Area | # videoQA | Reasoning | # videoQA |
|---|---|---|---|
| Memory | 7256 (36) | Descriptive | 31536 (106) |
| Abstraction | 12737 (58) | Explanatory | 4513 (14) |
| Physics | 23741 (80) | Predictive | 1278 (7) |
| Semantics | 24965 (82) | Counterfactual | 733 (5) |

Table 3: **Top**: Annotations in the *Perception Test*. Each object or point track contains frame-level annotations at a certain *frame rate*, *e.g.* each point is annotated on every frame, at 30 fps. Action and sound segments are annotated at the original video frame rate. # classes refers to the number of unique object names for object tracks and the number of unique questions for multiple-choice videoQA (mc-vQA) and grounded videoQA (g-vQA). **Bottom**: Number of videoQA pairs and (unique questions) per area and type of reasoning. Note that one question may be counted in multiple areas if it tests more than one skill. Each question is assigned a unique type of reasoning.

**Object tracks:** Object tracks represent the *root annotation* of our benchmark. All the other annotations, except for multiple-choice vQA, are linked or grounded into object tracks. In the annotation process, we instructed annotators to focus on the objects that the person interacts with and the objects that are in the immediate vicinity of the area where the person is performing actions, which act as distractor objects. We annotated boxes at 1fps throughout the video, which gives a good trade-off between density of annotations and annotation cost. When the objects are occluded, the annotators marked an approximate position of the boxes. Some ambiguous classes still remain, like liquids being poured or objects being torn. The object names were defined from an open vocabulary. The annotators typically included object attributes as well (colour, material), resulting in a large number of unique names. A list of the most frequent words (object or attributes) is included in the appendix, Fig. A1 (left), together with the distribution of object tracks into various categories, *e.g.* objects involved in actions or sounds correlated with camera motion (Table A1).

*Cups-game subset:* We isolate the videos corresponding to the cups-game scripts, as they can be an interesting subset for probing object trackers' abilities to reason about motion, object permanence, or occluded interactions when different factors may influence the difficulty of the task, *e.g.* identical vs non-identical objects used in the game, transparent vs non-transparent objects, or number of objects used. This subset contains 598 videos, with 483 videos where the cups are identical, and 113 videos where the cups are transparent. Most of the videos have 3 cups (451 videos), 132 videos have 2 cups, and 34 videos have 4 cups. We also provide a visibility mask for each video showing when the hidden object is occluded.

**Point tracks:** Although object tracks based on bounding boxes allow probing some physical properties of objects, such as object permanence, solidity, and coarse motion, they do not fully describe articulated or non-rigid objects, thin objects that are not axis-aligned, or out-of-plane rotation. A better understanding of physical interactions arises if we can track how object *surfaces* move and deform over time. To this end, we annotate point tracks on object surfaces following the protocol of TAP-Vid [19]. Annotators were instructed to select points spanning all the different parts of the objects labelled in the object tracking task. Points that are occluded are simply marked as occluded and not tracked. For translucent objects (*e.g.* glass cups), we only consider points to be 'visible' if they belong to the surface closest to the camera. The annotated points are dense in time (30fps). Table A2 in the appendix gives the distribution of points that are moving or static, as well as those on videos with moving cameras.

**Action segments with action-relevant objects:** To capture temporal understanding and enable grounding over time, we annotate the videos with temporal segments belonging to a fixed set of templated labels, *e.g. putting something into something*, similar to [26]. These are associated with action-relevant object tracks, i.e. objects involved in the action. The action boundaries are defined based on contact with action-relevant objects. For instance, when a person puts sugar in a tea, the *putting something into something* action starts when the person picks up the spoon and ends when the person puts down the spoon. If, after putting the sugar, the person starts stirring with the same spoon,

| Task | Output | Metric | Baseline | Score |
|------|--------|--------|----------|-------|
| Object tracking | box track | Avg. IoU | SiamFC [8] | 0.67 |
| Point tracking | point track | Avg. Jaccard | TAP-Net [19] | 0.40 |
| Temporal action localisation | list of action segments | mAP | ActionFormer [57] | 0.16 |
| Temporal sound localisation | list of sound segments | mAP | ActionFormer [57] | 0.15 |
| multiple-choice videoQA | answer (1 out of 3) | top-1 accuracy | SeViLA [55] | 0.46 |
| grounded videoQA | list of box tracks | HOTA [40] | MDETR [34]+Stark [52] | 0.10 |

Table 4: Computational tasks and top-performing baselines in the *Perception Test*: the model receives a video with audio, plus a task-specific input (*e.g.* the coordinates of a bounding box for the object tracking task), and produces a task-specific prediction, evaluated using dedicated metrics.

this defines a new segment as the type of action changed. The frequency of actions across the entire dataset is included in the appendix, Fig. A1 (right).

**Sound segments with sound-relevant objects:** Similarly to the action segment annotations but applied to the audio modality, we collect sound segment annotations grounded in object tracks. By watching the video and listening to the audio, the annotators define temporal sound segments and label them from a list of 16 audio segment labels. For each sound, the annotators also identify the object (or objects) involved in making the sound, or specify that these are out of the camera's field of view. For example, if an object is placed on the table making an audible sound, then both the object track and the table track are associated with the sound segment. The frequency of sounds across the entire dataset is included in the appendix, Fig. A2.

**Question-answers for video-level reasoning:** Different from the existing VQA datasets, which rely on crowd-sourced questions and answers, our team designed the questions per script to cover different types of reasoning [54]: descriptive, explanatory, predictive, counterfactual, and to cover aspects that are important for operating in the real world, *e.g.* understanding task completion, detecting changes, and so on. The answers for all the questions per video were provided by crowd-sourced participants. As we are interested in non-ambiguous evaluation, we favour the multiple-choice setup over the open-language answer setup. To define challenging negative options, we partly relied on human annotators, partly sampling from the correct answers of other videos in the same type of script. Table 3 bottom and Figure A3 in the appendix show the distribution of question-video pairs into perception skills, skill areas, and type of reasoning.

**Question-answers with answer-relevant objects:** As another way to connect high-level and low-level scene understanding capabilities, we define questions or tasks in language form, with answers given as object tracks. Similar to the multiple-choice question-answers above, our team defined the questions, and human raters selected the answers from the existing object tracks. The grounded questions are associated with skill areas and types of reasoning.

## 5 Computational tasks and baseline results in the *Perception Test*

**Computational tasks:** We defined six computational tasks based on the annotations available in the *Perception Test*. We summarise in Table 4 the task definitions (outputs, metrics) and the performance of top-performing baselines. It can be observed that the *Perception Test* combines lower-level dense prediction tasks like object and point tracking, whose outputs are box and point trajectories, with higher-level tasks like video question answering. For all the tasks, the video and audio are available as inputs, together with a task specification where applicable, *e.g.* the coordinates of a box to track for object tracking, or a language question and options for multiple-choice videoQA. More details about the task definitions are included in the appendix. Note that many other computational tasks can be defined based on the available annotations, *e.g.* grounded temporal action/sound localisation.

**Baselines:** Ideally, a single model should be able to perform all the tasks in the *Perception Test*. Since such a model is not available in the literature, we include results obtained with per-task baselines on the validation split for all the six tasks in the *Perception Test*; see Table 4 for a summary of top-performing baselines and their average performance, and the appendix for more details. When selecting these baselines, we favoured strong-performing models that can be evaluated in a zero-shot or few-shot setting, as our focus is on generalisation. However, for action and sound localisation, such models do not exist in the literature, so fine-tuning on our set of classes was necessary. For the mc-vQA task, we also provide a human baseline and fine-tuned evaluation to further assess the difficulty of the dataset for humans and for SOTA video-language models, respectively.

**Object tracking:** The overall performance of SiamFC [8] (UniTrack [50] implementation) on our benchmark confirms the findings from [21] that simple Siamese trackers are better when probed zero-shot than more complex recent trackers, *e.g.* Stark tracker [52] obtains 0.56 mean IoU on the *Perception Test* vs 0.67 for SiamFC. Even for SiamFC, the tracking performance drops when the camera and/or the objects are moving. The results for the different categories of objects (involved in actions or in sounds, etc.) are included in Table A3, aggregated based on camera motion.

**Point tracking:** The performance of our baseline TAP-Net [19] is a bit lower on the *Perception Test* compared to the performance reported by the authors on the Kinetics dataset [36] (0.466 vs 0.401); see detailed results in Table A4. We attribute this drop in performance mainly to the increased video length in our benchmark (23s in *Perception Test* compared to 10s in Kinetics).

**Action localisation:** The confusion matrix for our fine-tuned baseline ActionFormer [57] (Fig. A4) shows that the model struggles mostly with rare action classes that are confused with more frequent ones, and it also confuses pretend actions with their non-pretend versions, *e.g. ironing something* vs *pretending to iron something*. Using multimodal inputs does not increase the performance significantly, as summarised in Table A5, top. Overall, ActionFormer's performance on our benchmark is lower compared to other benchmarks (15.56 mAP on *Perception Test* vs 22.7 mAP on EPIC-Kitchens [14]), most likely due to the presence of adversarial actions and our limited training set. We hope to see in the near future models that can handle open-vocabulary action classes (similarly to open-vocabulary object detection [35]), so that fine-tuning is no longer necessary.

**Sound localisation:** We adapted the same ActionFormer model [57] to perform the localisation task in the audio modality. The best performance is obtained when features from both video and audio modalities are used as input; see Table A5, bottom.

**Multiple-choice vQA:** We report results for two strong recent video language models: Flamingo [3] in zero-shot and few-shot setups, and SeViLA [55] in zero-shot and fine-tuned regimes. We also include a dummy frequency-based baseline and a human baseline. For the frequency baseline, given that each question-options pair is defined over multiple videos, we keep as answer the option that is most frequently the correct answer in the training set. One can also compute this baseline on a random subset of training examples for each question, see Table A6, to obtain a fairer dummy baseline for models using few-shot evaluation.

*Human baseline.* We ran a small study for the mc-vQA task with human participants. We used 126 questions from the dataset, with one video per question selected at random. We recruited 30 crowd-sourced participants (half male, half female, with advanced English skills), different from the raters annotating the videos. Each participant answered a subset of 42 questions, resulting in 10 answers per question. The performance per area and type of reasoning is detailed in Figure 2. The overall average accuracy was 91.4%. The mistakes occurred in situations difficult to judge from the given viewpoint, *e.g.* if a configuration of objects would be stable (without seeing the end of the video), or in edge cases where humans overlooked details happening very early on in the video. It is worth noting that the participants did not require any training, which is similar to a zero-shot setup. The median time spent to answer 42 questions was 30 minutes.

It can be observed that both Flamingo and SeViLA are far from human performance when evaluated 0-shot or few-shot and cannot outperform the 8-shot dummy frequency baseline; see Figures 2, 3, A5, and Table A6. On many skills in the Memory, Physics, and Abstraction areas, their performance is below the 8-shot frequency dummy baseline, and in a few cases, *e.g.* (Piaget) conservation task, collision, or counterfactual reasoning, they are even below the pure random baseline. For counterfactuals, our qualitative investigation shows that Flamingo tends to latch on the visible elements in the video, failing to imagine the alternate reality that counterfactual questions require; *e.g.* for videos where a person writes some letters on the paper, the (counterfactual) question posed is: *What would be the order of the written letters if the person had written them in reverse order?*. Flamingo often selects the written order as correct. Interestingly, the larger versions of the model (due to larger language branches) seem to fare worse overall, which might point to overfitting issues. However, we leave an in-depth analysis for future work. Fine-tuning SeViLA leads to better results compared to all-shot frequency baseline, but mainly in the Semantics area (Fig. 3). SeViLA's 0-shot/fine-tuned scores on *Perception Test* (Fig. 3) are significantly lower than on NExT-QA benchmark [51]: 46.2 vs 63.6 for zero-shot, and 62.0 vs 73.8 on fine-tuned. We attribute this to the diversity of skills and the hard negative options included in the *Perception Test*.

**Grounded-vQA:** As no existing model can perform this task, we created a baseline by running MDETR [34] on the middle frame and then tracking the predictions using the Stark [52] object tracker. The performance is poor; see Table A7 and Figure A6. The failures are caused mainly by poor detection results – since the tasks are temporal in nature, extracting *seed* boxes from the middle frame is not sufficient to solve the tasks, calling for models capable of dealing with both spatial and temporal dimensions.

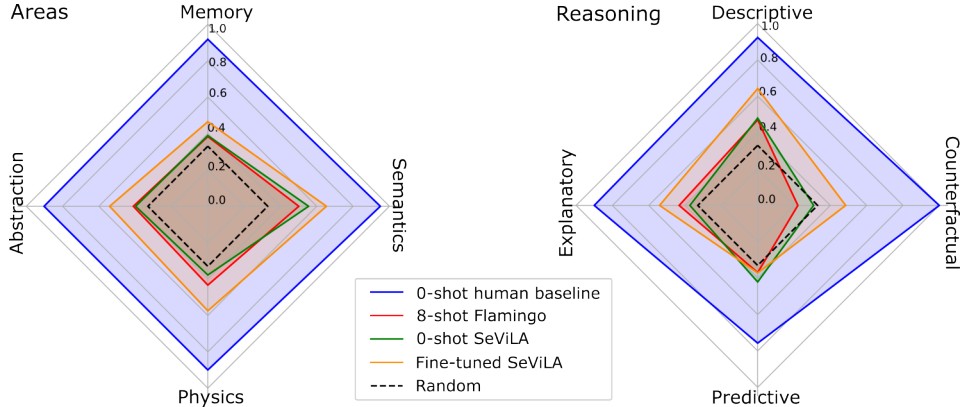

Figure 2: 0-shot human baseline compared to 8-shot Flamingo-3B, 0-shot and fine-tuned SeViLA, and random baseline on the validation set. In 0-shot and 8-shot regimes, both Flamingo and SeViLA are far from the 0-shot human baseline. SeViLA fine-tuning improves results to some extent, but the gap to human performance is still substantial.

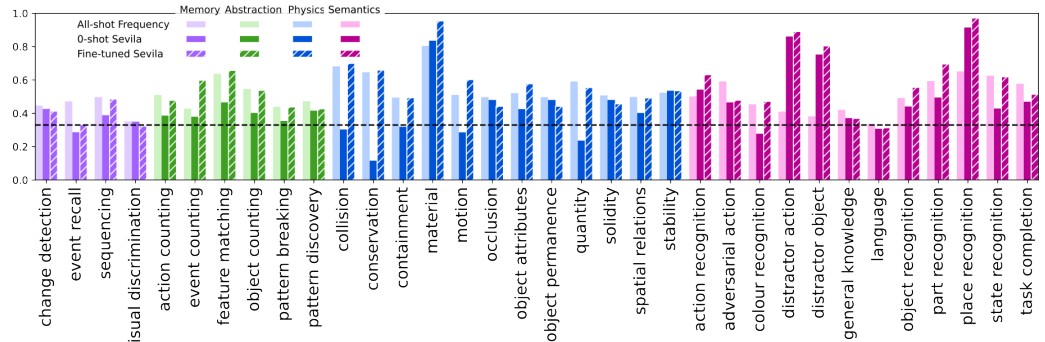

Figure 3: Performance on the validation set across skills for the 0-shot and fine-tuned SeViLA compared to frequency dummy baseline. The black dashed line indicates the random baseline.

## 6   Conclusion

We propose a diagnostic benchmark for multimodal models, to probe for memory, abstraction, physics, and semantic capabilities, across visual, audio, and text modalities, using real-world videos purposefully designed and filmed to show interesting perceptual situations. Solving the tasks requires different types of reasoning: descriptive, explanatory, predictive, and counterfactual. The videos are densely labeled with six types of annotations (objects and point tracks, action and sound segments, multiple-choice and grounded video question-answers). We are open-sourcing the videos and the annotations in the train and validation splits, together with per-task baseline results and evaluation code. A challenge server is available to evaluate models on the held-out test split. In principle, any model can be evaluated on our benchmark, either in a zero/few-shot setting or by fine-tuning on our limited train split. An ideal perception model would be able to perform all the tasks in our benchmark. Our results suggest that state-of-the-art zero-shot or few-shot video-language models are not able to outperform a dummy frequency-based baseline, whereas humans in the same setting are nearly perfect. We discuss limitations, and ethical and societal aspects in the appendix. We hope that our work will contribute to understanding models' limitations (through direct evaluation and interpretability analysis supported by the different types of annotations) and narrowing down areas of improvement to guide research towards general perception models.

## Acknowledgments

We are grateful to Luis Piloto, Kenneth Marino, Luyu Wang, Felix Hill, Martin Chadwick, Lucy Campbell-Gillingham, Boxi Wu, Drew Jaegle, Pauline Luc, Marianne Monteiro, Anna Bulanova, Radu Isac, Muqthar Mohammad, Vijay Vibha Tumala, Mahesh Maddinala, Yiwen Luo, Alina Kuznetsova, Aida Nematzadeh, Lisa Anne Hendricks, Aishwarya Agrawal, Nando de Freitas, Matt Botvinick, Shane Legg, and Relja Arandjelovic for providing insightful input on this project.

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
