# Appendix

## 1 *Perception Test* at a glance

Figure 1 and the presentation video available at `https://github.com/deepmind/perception_test` summarise the types of videos, annotations, and tasks available in the *Perception Test*.

## 2 More details about annotations in the *Perception Test*

The distributions of object and point tracks across camera motion and objects involved in actions, sounds, and grounded vQA are included in Table A1 and Table A2. Figures A1 and A2 present the frequency of popular words included in object names, and the distribution of actions and sounds respectively. Figure A3 shows the distribution of questions across skills.

| Camera | Static | Moving | Total |
|---|---|---|---|
| # total objects | 165552 | 26164 | 191716 |
| # action objects | 55344 | 6923 | 62267 |
| # sound objects | 56158 | 7666 | 63824 |
| # g-vQA boxes | 6795 | 2579 | 9374 |

Table A1: Object tracks involved in actions, sounds, and grounded-vQA, split by camera motion.

| Camera | Static | Moving | Total |
|---|---|---|---|
| # total points | 7791 | 783 | 8574 |
| # moving points | 3800 | 783 | 4583 |
| # static points | 3991 | 0 | 3991 |

Table A2: Point tracks available in the *Perception Test*, split by point and camera motion.

## 3 Computational tasks

**Single object tracking:** In this task, the model separately tracks every single object labelled in the dataset starting from an initial bounding box. In some cases ($\approx 20\%$) where the object is entering the field of view at the beginning or during the video, the first box may span only a few pixels, so it does not contain a representative view of the object. To deal with this problem, we use a heuristic to select a later frame, when the object is not touching the image boundary, to identify the query box for each object track. Performance is evaluated using the standard *average intersection-over-union* (IoU) metric, (also called average overlap), for evaluating long-term tracking without tracker re-initialization. It is defined as the average IoU over the entire track between the predicted and the ground-truth boxes [32, 10]. We also provide code for more fine-grained analysis, *e.g.* performance on objects in videos shot with static vs. moving cameras, objects involved in actions etc.

*Cups-game subset:* For the occluded object involved in cups-games, we use intersection as a metric for tracking (as opposed to Intersection-over-Union), to deal with the uncertainty of the position when the object is occluded.

**Single point tracking:** In this task, given a set of ground truth initial 2D point coordinates, the model should separately trace their spatial trajectories throughout the video. Performance is evaluated using the recently proposed *average Jaccard* metric for evaluating both long-term point tracking position and occlusion accuracy. This metric checks how similar the predicted and the ground-truth point tracks are, based on the average number of true positive matches, divided by the sum of true positives, false positives, and false negatives over the entire track [30, 19].

**Temporal action / sound localisation:** We define these two tasks similarly, as temporal segment detection problems. Given a video, the model predicts potentially overlapping temporal 1d-segment covering the actions/sounds and classifies them using a fixed set of labels. Performance is evaluated using the standard mean AP over classes [57] based on temporal IoU between predicted and ground truth temporal segments.

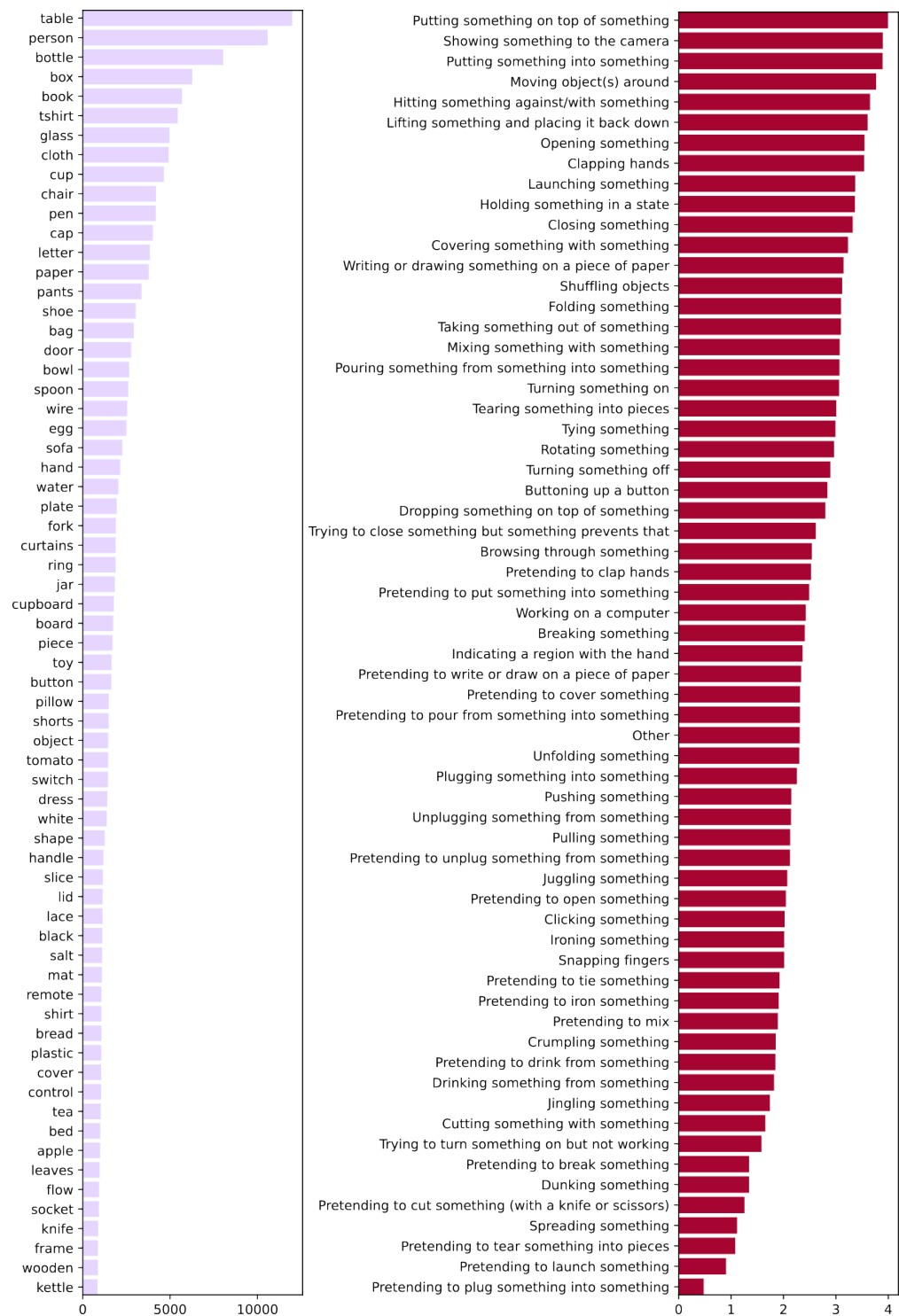

Figure A1: Frequency of objects and log-scale frequency of actions in the *Perception Test*.

**Multiple-choice video question-answering:** In this task, the model receives, in parallel with the video, a question and three possible answers, out of which only one is correct, and the model has to pick one answer (33% random chance). For most of the questions, watching the video and reading the question are enough for providing a correct answer. A limited number of questions are formulated in a generic way, so the options are necessary for choosing the answer: *e.g. Which of the*

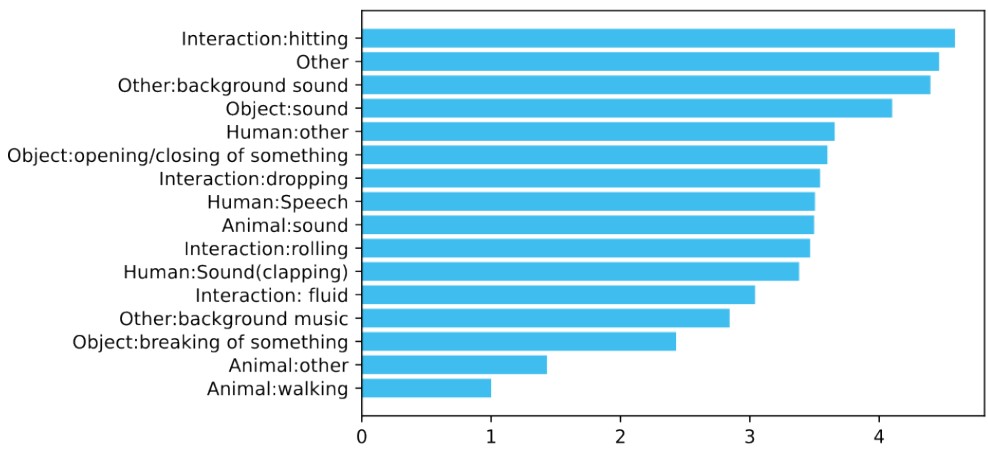

Figure A2: Log-scale frequency of sounds in the *Perception Test*.

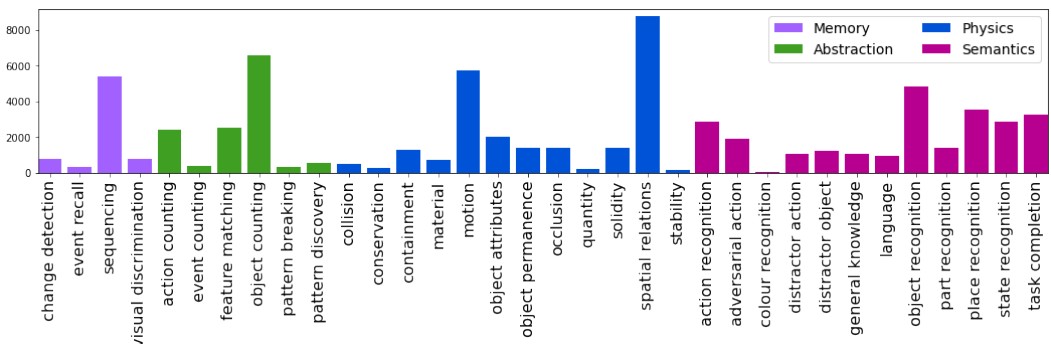

Figure A3: Number of multiple-choice video question-answers in the *Perception Test* across skills in the four skill areas: Memory, Abstraction, Physics, Semantics. One skill can be assigned to multiple skill areas—here we choose one as the prime area for each skill.

*following statements describes the scene better?*. In some cases, choosing the answer by elimination of the false options may be simpler. Performance is evaluated by measuring top-1 accuracy. For a couple of scripts, the videos must be trimmed to not reveal the answer: in the cups-games and stable configurations videos, we provide a frame id where the video should be trimmed. For the train and validation splits we release the entire videos together with the cut frame id information. In the held-out test split, only the trimmed videos are available for these particular video types.

**Grounded video question-answering:** This task is similar to conditional multiple-object tracking, with the conditioning given as a language task or question as opposed to a class label [38]. The answers are object tracks defined throughout the video and we use HOTA [40] metrics to evaluate performance. In some situations, the initial parts of the track might not be relevant for the question, *e.g. Track the object that was removed from the table* and the object is removed halfway through the video. However, given that we do not enforce causal processing of the video, the track prediction for the initial part can still be done in hindsight.

## 4 Baseline results

**Object tracking:** We report baseline results using the SiamFC model [8] (UniTrack [50] implementation). SiamFC was chosen due to its high-performance on a number of single-object tracking benchmarks when running in zero-shot setting [21]. We also include a static dummy baseline that

assumes all objects are static, so it just replicates throughout the video the box-to-track received as input. The results for the different categories of objects (involved in actions or in sounds, etc) are included in Table A3, aggregated based on camera motion.

| Object Tracking | All | Static camera | Moving camera |
|---|---|---|---|
| **all objects** | 0.66 / 0.67 | 0.70 / 0.69 | 0.42 / 0.54 |
| **action objects** | 0.48 / 0.53 | 0.50 / 0.54 | 0.31 / 0.47 |
| **sound objects** | 0.56 / 0.60 | 0.58 / 0.61 | 0.40 / 0.53 |
| **g-vQA boxes** | 0.38 / 0.50 | 0.43 / 0.51 | 0.26 / 0.47 |

Table A3: Static dummy baseline / SiamFC results, measured as average IoU, across different categories of objects in the *Perception Test*. Since many objects are static, the performance of the dummy baseline is good overall, but it degrades considerably when motion is involved, whereas the SiamFC tracker is more robust.

**Point tracking:** We report baseline results using a TAP-Net model [19] trained on Kubric [30] and transferred zero-shot. The model operates on 256x256 resolution (aspect ratio is not preserved) and consumes the whole video directly. We also include a static dummy baseline assuming all future points are visible and never change the location. Table A4 shows the results. As expected, both moving points and points seen through a moving camera are considerably harder to track. Following [19], we use three evaluation metrics. (1) *Position Accuracy ($< \delta^x$)*: for a given threshold $\delta$, we measure the fraction of points that are within the threshold of their ground truth, for frames where points are visible. For all predictions, we resize them to 256x256 resolution and measure $< \delta^x$ across 5 thresholds: 1, 2, 4, 8, and 16 pixels. (2) *Occlusion Accuracy (OA)*: a simple classification accuracy for the point occlusion prediction on each frame. (3) *Jaccard at $\delta$*: an evaluation metric considering both occlusion and position accuracy. It is the fraction of 'true positives', i.e., points within the threshold of any visible ground truth points, divided by 'true positives' plus 'false positives' (points that are predicted visible, but the ground truth is either occluded or farther than the threshold) plus 'false negatives' (groundtruth visible points that are predicted as occluded or the prediction is farther than the threshold). Our final metric *Average Jaccard (AJ)* averages Jaccard across all 5 thresholds: 1, 2, 4, 8, and 16 pixels.

To further understand the performance, we split points into two groups: static and moving. Note that there are no static points in the moving camera scenario, all points are moving. In static camera, we determine that a point is moving if its distance between start frame and end frame is more than 0.01 in the normalized image coordinate system. As expected, the dummy baseline performs well on static points, reaching 0.722 average jaccard. But TAP-Net significantly outperforms when points are moving, particularly in the moving camera setup, improving average Jaccard from 0.088 to 0.328. Besides AJ, TAP-Net significantly improves the static baseline on occlusion accuracy from 0.675 to 0.849. One interesting observation is that on both position accuracy ($< \delta^x$) and jaccard at $\delta$, TAP-Net starts to outperform static baseline only when measured above 4 pixel threshold. This is because human annotations still contain small localization errors and 4 pixel threshold is more reliable than 1 or 2 pixel threshold for measuring under 256x256 resolution.

**Temporal action localisation:** We obtained baseline results for temporal action localisation using ActionFormer [57] with different pretrained features: TSP video features from [5] extracted using a Resnet(2+1)D-34 model pre-trained on ActivityNet, MMV audio features from [2] extracted using an S3D model pre-trained on AudioSet, and a multimodal input by concatenating the video and audio features. The video features have 512-dim and an effective stride of 32 (corresponding roughly to one feature per second): every other input frame is skipped and the model performs a temporal downsampling of 16. The audio features are extracted using a window length of 960ms, window stride 16000. The input audio is downsampled from 48khz to 16khz (keeping every third sample). This results in roughly 2 features per second, each of dimension 256. When using multimodal inputs, the video features are tiled over time (factor 2) to align them with the audio features.

We trained the transformer blocks and the classification and regression heads to accommodate for the number of classes included in our dataset. The resulting mean average precision is included in Table A5, top. The baseline struggles mostly with rare action classes and pretend actions, which are confused with their non-pretend counterpart class. Using only the audio modality leads to very poor performance, whereas using multimodal inputs does not increase the performance significantly.

| Point tracking | All points | static points static camera | moving points static camera | moving points moving camera |
|---|---|---|---|---|
| static baseline | 0.361 | 0.722 | 0.373 | 0.088 |
| TAP-Net [19] | 0.401 | 0.496 | 0.399 | 0.328 |

| Point tracking | OA | $< \delta^0$ | $< \delta^1$ | $< \delta^2$ | $< \delta^3$ | $< \delta^4$ |
|---|---|---|---|---|---|---|
| static baseline | 0.675 | 0.395 | 0.512 | 0.601 | 0.695 | 0.784 |
| TAP-Net [19] | 0.849 | 0.055 | 0.214 | 0.687 | 0.927 | 0.956 |

| Point tracking | Jac. $\delta^0$ | Jac. $\delta^1$ | Jac. $\delta^2$ | Jac. $\delta^3$ | Jac. $\delta^4$ |
|---|---|---|---|---|---|
| static baseline | 0.217 | 0.301 | 0.364 | 0.429 | 0.495 |
| TAP-Net [19] | 0.025 | 0.104 | 0.442 | 0.699 | 0.734 |

Table A4: Static baseline vs TAP-Net results on the validation set. **Top**: Average Jaccard (AJ), higher is better. There are no static points in the moving camera scenario. **Middle**: Occlusion Accuracy (OA) and Position Accuracy ($< \delta^x$), higher is better. TAP-Net outperforms static baseline when measured above 4 pixel threshold. **Bottom**: Jaccard at $\delta$, higher is better. TAP-Net outperforms static baseline when measured above 4 pixel threshold.

Figure A4 shows the confusion matrix for the action localisation task, normalised by columns. It can be observed that the less frequent actions are often confused with more frequent ones and the model also confuses pretend actions with their non-pretend versions, *e.g. ironing something* vs *pretending to iron something* or *writing or drawing something* vs *pretending to write or draw*.

| Model | Modality | Temporal Action Localisation | | | | | | |
|---|---|---|---|---|---|---|---|---|
| | | @0.1 | @0.2 | @0.3 | @0.4 | @0.5 | Avg | # epochs |
| **ActionFormer** | video | 17.67 | 16.56 | 15.13 | 13.28 | 11.07 | 14.74 | 35 |
| **ActionFormer** | audio | 7.25 | 6.53 | 5.70 | 4.67 | 3.64 | 5.56 | 55 |
| **ActionFormer** | video+audio | 18.82 | 17.63 | 15.98 | 13.99 | 11.37 | 15.56 | 35 |

| Model | Modality | Temporal Sound Localisation | | | | | | |
|---|---|---|---|---|---|---|---|---|
| | | @0.1 | @0.2 | @0.3 | @0.4 | @0.5 | Avg | # epochs |
| **ActionFormer** | video | 17.85 | 15.54 | 13.81 | 12.11 | 5.89 | 13.04 | 55 |
| **ActionFormer** | audio | 16.28 | 13.58 | 10.80 | 8.43 | 5.87 | 10.99 | 80 |
| **ActionFormer** | video+audio | 22.24 | 18.99 | 15.36 | 11.99 | 8.74 | 15.46 | 55 |

Table A5: Mean average precision (mAP) for temporal action localisation (top) and sound localisation (bottom) tasks using ActionFormer as baseline. IoU for 0.1-0.5 are averaged as in [15]. # epochs represents the number of training epochs used to obtain the best results for each experiment setup.

**Temporal sound localisation:** We use the same model architecture and pre-trained features as above. We trained from scratch the transformer blocks and the classification and regression heads. For both training and evaluation, we keep only 11 sound classes, excluding the classes corresponding to indistinguishable sounds (*e.g. Other:background*, *Other:human*), as they hinder learning. The resulting mean average precision is included in Table A5, bottom. The best performance is obtained when features from both modalities are used as input.

**Multiple-choice videoQA:** For this task, we provide results for two strong recent video language models: Flamingo [3] in zero-shot and few-shot setups, and Sevila [55] in zero-shot and fine-tuned regimes. We also include a dummy frequency-based baseline and a human baseline; see Table A6, Figure 2 and A5.

*Frequency baseline.* Given that we have a fixed set of question-answer pairs defined over multiple videos, we define a simple baseline that computes how frequently each of the three options is the correct answer in the training set, and keeps the most frequent one. This baseline obtains 55.1%. One can also compute this baseline on a random subset of training examples for each question type, see Table A6. This is a fairer dummy baseline for models using few-shot evaluation. Note that the dummy random baseline is equivalent to a 0-shot frequency baseline. The fact that the 8-shot performance of this dummy frequency baseline is above the 0-shot (random) baseline performance indicates an imbalance between the frequencies of correct answers across options in the dataset. This happens mostly because a number of questions in the dataset are binary in nature (*e.g. Is the camera moving or static?*), but a third option was added to comply with the 3-options-per-question setting; so in this

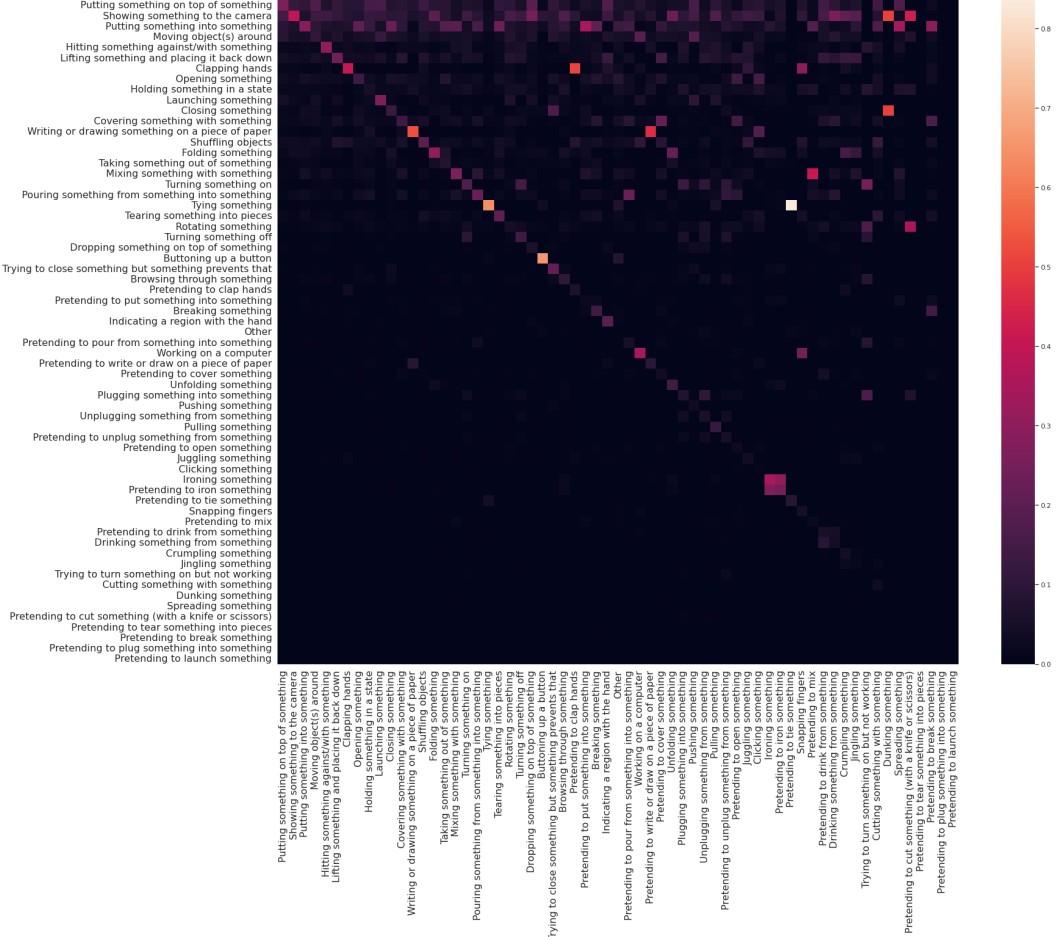

Figure A4: Confusion matrix for ActionFormer predictions on the action localisation task. To be considered as a prediction for a certain segment, the model's confidence has to be above 0.1 and IoU threshold between the prediction and ground truth above 0.1. Ground truth actions are listed on the y-axis, sorted by their frequency; entries are normalised by rows. The less frequent actions are often confused with more frequent actions. The model also confuses pretend actions with their non-pretend versions, *e.g. ironing something* vs *pretending to iron something* or *writing or drawing something* vs *pretending to write or draw*.

case the possible options are: (1) *moving*, (2) *static or slightly shaking*, (3) *I don't know*, but the third option is never or very rarely the correct one, bringing the performance of this dummy frequency baseline slightly above 50%.

*Flamingo*. We run the model with a maximum of 30 frames sampled at 1fps, spatial resolution 320. When the videos are longer than 30 seconds, we use only the middle clip. The audio modality is ignored as the original model was not trained to deal with it. The different options are scored based on likelihood. We considered zero-shot and 8-shot settings; see results in Table A6. In the zero-shot setting, the smaller version of the model obtains 43.6% on the test set. In the 8-shot setting, we sample 8 examples and associated ground truth responses from each question in the training set and use as prompts. The resulting accuracy is 45.8%, again obtained by the smaller version of the model. The performance per skills is detailed in A5

*SeViLA*. We run zero-shot and fine-tuned evaluation for the SeViLA model using the scripts provided by the authors [55]. SeViLA has a Localizer and an Answerer module. It starts by sampling uniformly 32 frames from the video, out of which 4 frames are designated by the Localizer as keyframes that the Answerer uses for the final prediction. When fine-tuning, we update the weights of both the Localizer and Answerer modules using the training set from the *Perception Test*. Fine-tuning improves

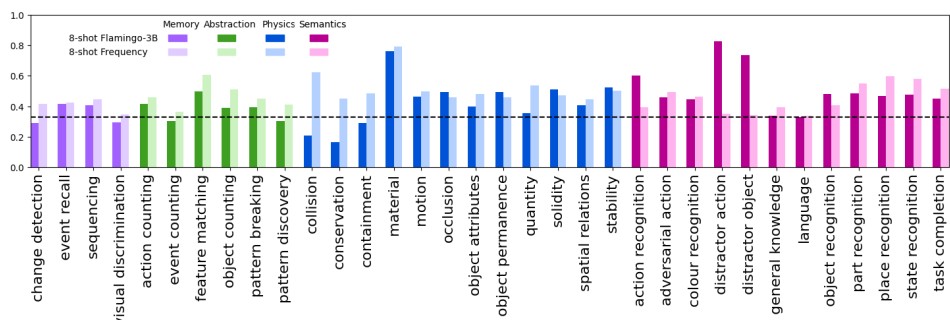

Figure A5: Performance on the validation set across skills for the 8-shot Flamingo-3B compared to 8-shot dummy frequency baseline. The black dashed line indicates the random baseline.

performance (from 46.2 zero-shot to 62.0), but this is still far from 0-shot human performance (91.4). The performance per skills is detailed in 3.

| mc-vQA | 0-shot | 8-shot | All-shot | Fine-tuned |
|---|---|---|---|---|
| **Flamingo-3B** | 43.6 | 45.8 | - | - |
| **Flamingo-9B** | 40.5 | 44.4 | - | - |
| **Flamingo-80B** | 41.6 | 45.4 | - | - |
| **SeViLA** | 46.2 | - | - | **62.0** |
| **Frequency** | 33.3 | **51.0** | **55.1** | - |
| **Human** | **91.4** | - | - | - |

Table A6: mc-vQA top-1 accuracy (higher is better), for different evaluation modes and different models, including a human baseline, on the validation split. "-" refers to numbers that were not collected.

**Grounded video question-answering:** In absence of a dedicated baseline in the literature for the type of grounded videoQA that we propose (input: text query, output/answer: object tracks), we obtain a simple baseline by running MDETR [34] on the middle frame of each video using the query as input, and then we use Stark tracker [52] to propagate the MDETR detections forward and backward in the video; we tried using SiamFC as tracker, but the results were worse. We measure the performance of this baseline using HOTA metrics, which integrate detection, association, and localisation scores. As expected, the performance of this baseline is poor; see Table A7 and Figure A6. The failures are caused mainly by poor detection results – since the tasks are temporal in nature, extracting *seed* boxes from the middle frame is not sufficient to solve the tasks.

| Model | HOTA | LocA | DetA | AssA |
|---|---|---|---|---|
| **MDETR+Stark** | 0.1 | 0.68 | 0.03 | 0.33 |

Table A7: HOTA results on the validation split for the grounded vQA task in the *Perception Test*.

**Model size** is an essential aspect that impacts real-world applications. We report in Table A8 the number of parameters of the evaluated models. For more details about the training cost of these models or inference speed, we refer to the original papers introducing these models.

## 5 Dataset Splits Generation

The 11.6k videos in the *Perception Test* are split into train, validation, and held-out test splits each with roughly $20\%/50\%/30\%$ of the videos respectively. These splits were generated by respecting two constraints: (1) all videos from each unique combination of (`script_id`, `participant_id`) are kept in the same split; more specifically, each script was filmed by a given participant possibly with multiple camera configurations, *e.g.* from different viewpoints, or both with static and moving cameras. The above constraint ensures that all such variations of a script shot by a participant belong in the same split to avoid any leakage of video content across splits, and (2) various video attributes (camera motion, indoor *vs.* outdoor) and annotations are divided in the same proportion across splits, *e.g.* each split will have approximately the above specified fraction of videos with moving camera, or

| Model | Task | # params |
|---|---|---|
| SiamFC | object tracking | 25.6M |
| TAP-Net | point tracking | 2.8M |
| ActionFormer-action | temporal action localisation | 27.0M |
| ActionFormer-sound | temporal sound localisation | 26.5M |
| Flamingo | multiple-choice videoQA | 3B |
| SeViLA | multiple-choice videoQA | 4.1B |
| MDETR+Stark | grounded videoQA | 209M |

Table A8: Number of parameters of models evaluated on our benchmark.

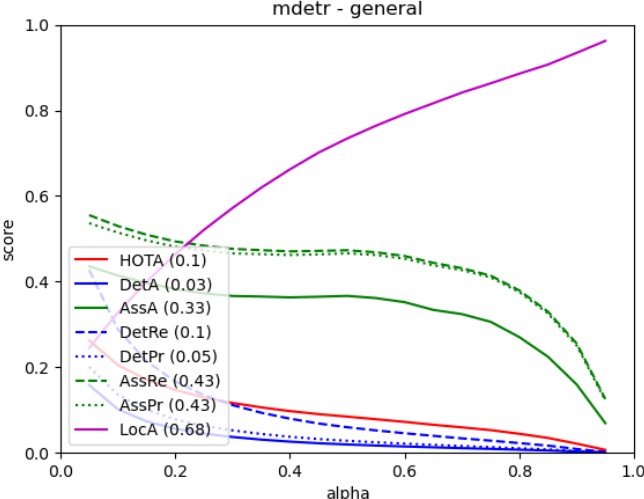

Figure A6: HOTA metrics for MDETR+Stark tracker baseline on the validation split of the *Perception Test*.

with point annotations. In particular, each question in the multiple-choice and grounded video QA tasks applies to a number of videos; this constraint ensures that these videos are distributed across splits in the specified proportion, such that all questions are present in all the splits.

The above was executed by setting up a linear program with a binary decision variable for each unique (`script_id`, `participant_id`) pair indicating which of the two splits it should be assigned to, denoted collectively $\mathbf{x} \in \{0,1\}^n$ with $n$ being the number of such unique pairs. Note for splitting into three splits, the problem is solved twice sequentially. A feature count matrix $A \in \mathbb{R}^{n \times d}$ was constructed, with $A_{ji}$ being the number of videos shot by the $j^{th}$ (`script_id`, `participant_id`) having the $i^{th}$ video-attribute ($d$ being the total number of video attributes). An "attribute" indicating the total number of videos with a given (`script_id`, `participant_id`) was also included to enforce the number of videos in each split. The following linear program was solved using the CVXPY interface to the MOSEK mixed-integer solver.

$$\min_{\mathbf{x}} \left[ \left( \max_i (1 - t_i)^2 \right) + \frac{1}{d} \sum_{i=1}^{d} (1 - t_i)^2 \right]$$

$$\text{s.t.,} \quad t_i = \frac{A_i^T \mathbf{x}}{\lceil f_1 A_i^T \mathbb{1} \rceil}, \forall i \in \{1, \ldots, d\}$$

$$(1 - \lambda) \le t_i \le (1 + \lambda), \forall i \in \{1, \ldots, d\}$$

$$\text{and,} \quad \mathbf{x}_j \in \{0, 1\}, \forall j \in \{1, \ldots, n\}$$

with $A_i$ being the $i^{th}$ column of $A$, $f_1 \in [0, 1]$ being the target fraction for the split corresponding to label $\mathbf{x}_j = 1$ (*e.g.* $f_1 = 0.5$ for a 50% test split), and $\lambda = 0.25$ is the maximum allowed fractional deviation from the target value. There were $n = 7288$ unique (`script_id`, `participant_id`) pairs, and $d = 249$ video attributes.

# 6 Annotation collection

**Raters instructions:** We include below the high-level instructions provided to raters when collecting the different types of annotations.

*Object tracks:* Annotate, using spatio-temporal bounding box tracks, all the objects that the person interacts with. Annotate also the objects in the immediate vicinity of the objects that the person interacts with, as they act as distractor objects. In addition, annotate 3-5 objects in the background. Once you mark an object in a frame, the tracker running in the background will generate proposals for that object throughout the video. Please check every 30th frame and amend the proposals if they are not correct. For each object track, provide a representative name including object class, object attributes, *e.g.* red mug. As much as possible, include annotations for liquids as well. When objects get torn (*e.g.* a salad leave, a piece of paper) or are broken (*e.g.* eggs), the object tracks and names should reflect the change in object state: *e.g.* a single object track named "egg" would be split into multiple object tracks named "egg-shells" and "egg-content" once the person breaks the egg. For objects that go out of the field of view and reappear later on, make sure to assign the same object track.

*Point tracks:* Given a video with an inpainted box track, select and track at least 3 points inside the object, belonging to different parts of the object. Mark the start and end points of the track, then wait for the optical flow estimator running in the background to provide predictions for the intermediate frames. Check and correct any errors you notice on all the intermediate frames. Assign names to each point corresponding to the semantic part to which the point belongs to. When a point becomes occluded (because of object rotation or object going out of the field of view), mark the point as *occluded*.

*Action segments:* Given the list of templated action labels (*e.g.* putting *something* into *something*), mark the start and end points of each action segment. As action boundaries, please use the moments of contact with the objects involved in the action, *e.g.* when a person stirs a tea with a spoon, mark as start moment the moment when the person picks up the spoon, and as end moment the moment when the person stops stirring or puts down the spoon. In addition to indicating the action boundaries, select from the existing object tracks the tracks involved in the action segment, in the order in which they appear in the template, *e.g.* when a person pours water from a kettle into a cup, mark the segment as *pouring something from something into something*, and indicate *water*, *kettle*, *cup* as relevant objects, in this order. If the person repeats the same action multiple times (*e.g.* clapping hands), mark separate segments as much as possible.

*Sound segments:* Given the list of sound labels (*e.g.* clapping hands, hitting something), mark the start and end points of each sound segment. In addition, select from the existing object tracks the tracks involved in producing the sound, *e.g.* when the person drops a cable on a table, mark the sound as *hitting something*, and select *cable*, *table* as objects involved in producing the sound.

*Multiple-choice videoQA: Phase 1* (open-ended questions): Answer the following questions about this video using short sentences written in English. *Phase 2* (multiple-choice questions): Answer the following questions about this video by selecting the correct answers from the given ones. Only one option is correct for each question.

*Grounded videoQA:* The answers to questions were generated automatically from the object tracks annotations above, using simple heuristics. These annotations were then checked by human raters with the instruction: Given the question or query below and the video with one or more inpainted object tracks, indicate (yes/no) if the inpainted objects correctly answer the question.

**Data collection pipelines:** The different types of annotations were collected using two different approaches:

1. *sequential pipeline* for the object and point tracks, action and sound segments: (i) a rater annotates a video for a given task, (ii) a second rater checks the annotation, makes any necessary corrections, then marks the annotation as complete; (iii) a third rater checks if the annotation is indeed complete or it needs additional changes, in which case they will send the video back to step (ii) to be reviewed by a different rater. For difficult tasks like point tracking or object tracking with hard occlusions, we did multiple annotation cleaning rounds, each time with specific cleaning guidelines. For example, for the videos in cups-games category mentioned above, in one cleaning round, the raters were asked to pay attention to

the hidden object, or for videos where the person shows objects to the camera sometimes repeating the same object, we asked raters to pay attention to assign the same object ID when the object reappears. Having videos grouped by script type helped in designing specific cleaning guidelines to ensure good annotation quality.

2. *parallel pipeline* for multiple-choice and grounded videoQA: multiple raters answer in parallel the same question for the same video and the option chosen by the majority of raters is kept as final answer. Note that for multiple choice QA, during annotation collection, the raters were presented with more than 3 options in some cases. For the final dataset, as the goal was to have the same number of options for all the questions, we chose to keep 3 options to accommodate binary questions as well (where the options used are: *Yes*, *No*, *I don't know*). For questions with more than 5 options, the negative options were sampled based on their frequency as correct options for videos in the same script type. Finally, for some generic questions, *e.g. Which statement describes the scene better?*, the answers were collected initially in open-language format, and then negatives were sampled using the answers from other videos in the same script type, with additional checks from the research team to avoid ambiguous distractors.

As a sanity check, for the action and sound annotations, we checked for overlapping objects involved in both action and sounds (see Figure A7). We observed strong correlations across pairs of action-sound, indicating consistent annotations across modalities, *e.g.* the *Pouring something into something* action shares the same objects with the *Interaction: Fluid* sound, the *Clapping hands* action co-occurs with the *Human (clapping)* sound, the *Lifting something and putting it back down* action co-occurs with the *Object: Hitting* sound, *Moving something around* actions co-occurs with *Object: Rolling* sound, and so on.

# 7    Compensation

Besides the research team creating the benchmark, we relied on three groups of contributors: participants filming the videos, raters annotating the videos, and human participants involved in the human baseline collected for the multiple-choice vQA task. The full details of our study design, including compensation rates, were reviewed by DeepMind's independent ethical review committee. All participants provided informed consent prior to completing tasks and were reimbursed for their time. The policy ensures that workers/participants are paid at least the living wage for their location.

# 8    Diversity of participants involved in filming

We consider that good representation of the world's population in terms of different demographics is an essential aspect in benchmarking multimodal models, to ensure a safe and fair deployment of such models world-wide. When building our benchmark, we considered three diversity aspects for the participants involved in filming: gender, ethnicity, and country of residence. These factors offer visual diversity in the dataset in terms of appearance of people and scenes. We acknowledge that this is not a complete coverage of diversity factors, and other aspects such as age, disability, household income, or level of education are important to control, and we hope to be able to include such factors in future iterations of the benchmark.

We include in Table A9 and Figure A8 the self-reported demographics along these axes. Note that all the demographic information was self-reported by the participants themselves. It can be observed that there is a good balance across gender and good spread across ethnicity (providing diversity in terms of skin-tone). Filming the videos in more than 13 countries on different continents provides good scene and objects variation.

# 9    Limitations and potential negative societal impact

**Limitations:** We designed video scripts and questions to have a broad coverage of perception skills and types of reasoning, across different modalities (video, audio, text), probed through high-level and low-level computational tasks. Given the broad coverage, it was challenging to have a perfect balance across all dimensions. As future work, we aim to add more tasks that require counterfactual

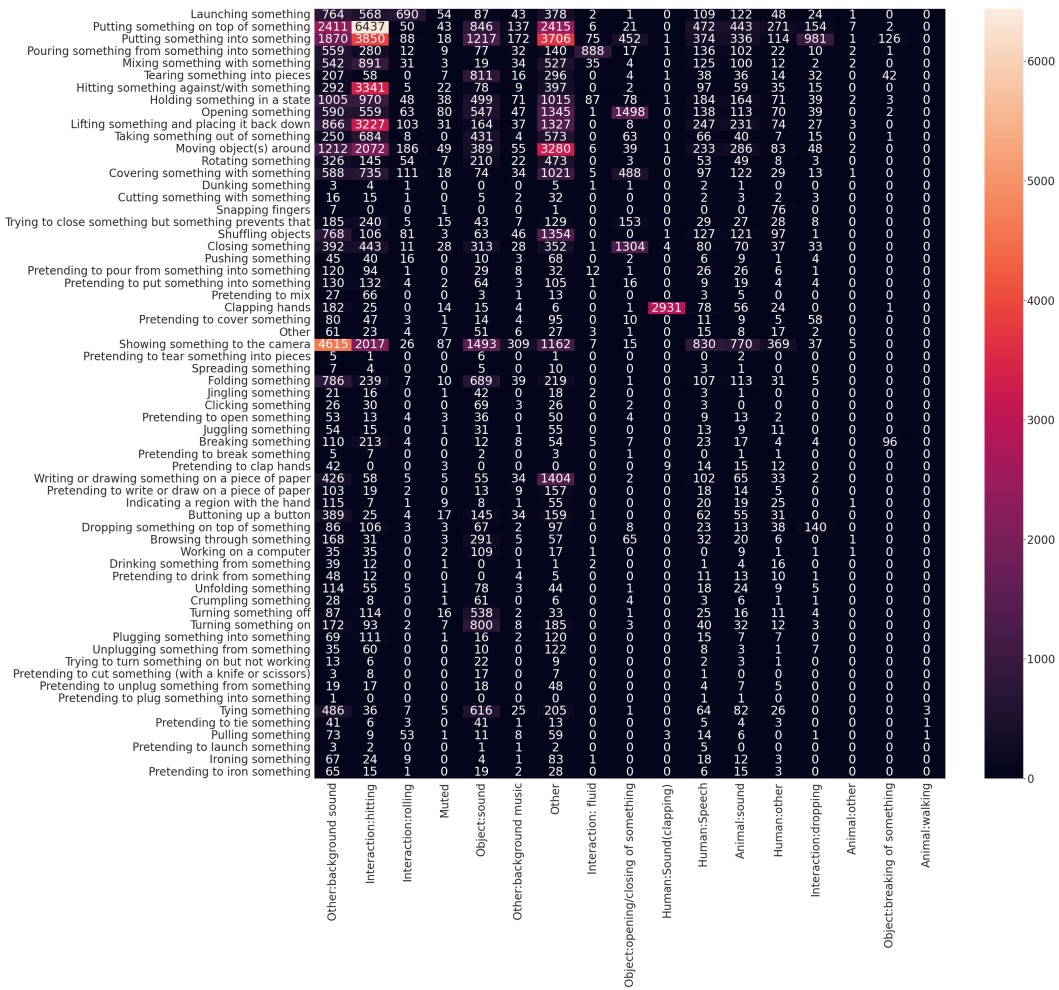

Figure A7: Correlation between action and sound temporal annotations in the *Perception Test*.

reasoning or memory skills, and more annotations for grounded vQA and point tracking. In addition, the balance across options in the multiple-choice videoQA is not perfect, as indicated by the fact that the frequency dummy baseline obtains better performance compared to the pure random baseline (55.1% vs 33.3%). When analysing model performance, we need to take such biases into account.

Our benchmark aims to comprehensively evaluate multimodal models' performance across different perception skills. However, some modalities are missing, *e.g.* touch, or some aspects are not covered by the available annotations, *e.g.* force, deformations, detailed 3D geometry. We will work to improve the coverage in future iterations, and we also welcome contributions from the community to add more tasks, modalities, even new languages to the *Perception Test*. Note, however, that our benchmark focuses on temporal tasks defined over videos. Many current multimodal models can only handle image and text as modalities, hence it might not be straightforward to evaluate them on our benchmark. We do not consider this to be a limitation of our benchmark, but a limitation of those models, as we believe that general multimodal perception models need to be able to perceive and reason over spatial and temporal dimensions, across modalities.

As mentioned in the main paper, most of the videos were filmed on a table-top, using common household objects, following the scripts designed by our research team. This could be perceived as a setup with limited diversity when compared to "in the wild" videos available in repositories like Youtube. However, we argue that for evaluating a general perception model, it is important to isolate the skills and types of reasoning we care about, while building in invariance to lighting, camera angle, types of objects, the person's skin tone, etc – these are obtained by filming the same script (with multiple variations) with 20+ different participants per script variation, who choose on their own

| Gender | % |
| --- | --- |
| Male | 46.40 |
| Female, Other | 53.60 |

| Ethnicity | % |
| --- | --- |
| White or Caucasian | 28.97 |
| South and East Asian | 25.49 |
| Black or African American | 21.68 |
| Latino or Hispanic | 9.25 |
| Mixed | 3.94 |
| Middle Eastern | 3.37 |
| Other | 7.30 |

| Country | % |
| --- | --- |
| Philippines | 31.38 |
| Brazil | 11.27 |
| Kenya | 10.02 |
| Indonesia | 8.75 |
| Italy | 8.03 |
| Romania | 7.57 |
| South Africa | 5.25 |
| Turkey | 4.12 |
| India | 3.72 |
| Mexico | 1.45 |
| Bulgaria | 1.37 |
| United States | 0.70 |
| Egypt | 0.48 |
| Other | 5.87 |

Table A9: Self-reported demographics (Gender, Ethnicity, Country) of participants involved in filming.

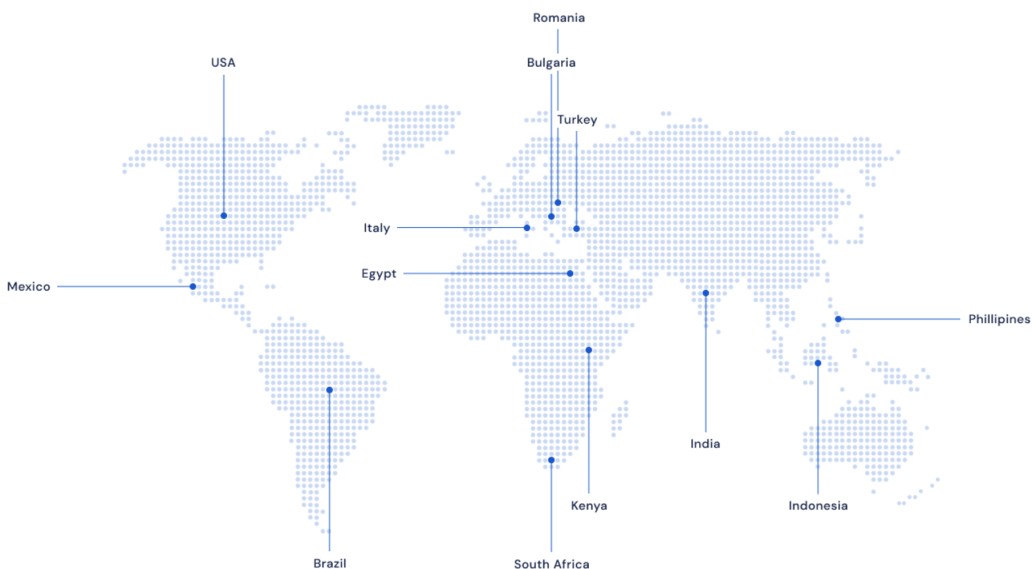

Figure A8: Geolocation of participants involved in filming.

where to place the camera, what exact type of object to use for a certain action, in what order to perform some actions, etc. Curating videos in the wild to obtain the same coverage of skills and types of reasoning would be hard, even impossible, since some types of data simply don't exist online in sufficient numbers (*e.g.* correct vs. incorrect execution of actions).

While filming, we instructed participants to use 2 different viewpoints to obtain more diverse camera angles. However, this information is not explicitly used at the moment in our tasks (we only used this information when deciding the splits, to make sure these paired videos fall in the same split). While some participants filmed simultaneously with two cameras, others recorded two runs one after the other. This approach was previously used in crowd-sourced datasets (e.g. Charades-Ego dataset). Through manual inspection, we note that the variations are minor as the sequences were recorded one after the other. Such paired videos could be useful to design new tasks.

Related to missing capabilities, our benchmark cannot accommodate *active* perception evaluation. Enabling interaction would limit us to simulated environments. To still address the agency aspect to some extent, we included videos where the model is required to recognise correct and incorrect execution of certain actions (e.g. tying shoe laces, buttoning up a shirt, covering a container with a cover, pouring water in a glass) or to assess the consequences of actions (e.g. what would happen if we remove a certain object from the table) – these are possible because our scripts include multiple variations with correct/incorrect actions, or controlled variations of object configurations, which would be impossible to curate from public repositories like Youtube.

As mentioned in the main paper, our benchmark is medium scale, comparable to Charades, but an order of magnitude smaller compared to *e.g.* Ego4D. We would like to emphasise that this is not a limitation of the benchmark since our focus is on evaluation; large-scale datasets are needed for training. We provide a medium-scale dataset with a small set for fine-tuning / prompting, and the rest for evaluation, as in this way we can probe the generalisation power of multimodal (pre-trained) models.

**Societal impact:** Benchmarks such as ours guide research and, indirectly, lead to improvement in models' capabilities. These models can be used for many different applications that have positive societal impact, *e.g.* video-language models assisting the visually-impaired, or surveillance systems for the elderly or children. However, similar systems can be used to cause harm (*e.g.* intrusive surveillance systems). We hereby state our strong stance against the use of our benchmark for evaluating and improving such systems.