# OpenReview forum: "Perception Test: A Diagnostic Benchmark for Multimodal Video Models"
_NeurIPS.cc/2023/Track/Datasets_and_Benchmarks — NeurIPS 2023 Datasets and Benchmarks Poster_

### Official Review · Reviewer_mVd6 · 2023-07-07
**Interesting Dataset, but does not motivate its utility due to limited analysis and insights provided**

**Rating:** 5
**Confidence:** 5
**Correctness:** I do not see any issues with correctn…

**Strengths:**

**Summary**
I think the dataset is clear in its motivation and collection process. I also think the different aspects of the dataset are very useful to better understand where models are strong and weak and what areas of improvements can be made.

**Details**
* When filming videos, they ensured ethnic variety to prevent undesirable biases.

* The details on the demographics (location, ethnicity, camera, etc) were provided.

* Table 2 provides good clarity on the scripts discussed.

* The supplementary provides details on the models that were evaluated on the benchmark.

* On some tasks they provided human results, giving us another baseline to compare to that is useful for future work. They also provide a frequency-based baseline, allowing for further comparisons.

* In supp Section 5, there is a detailed explanation of how the datasets were split to ensure less bias in the distribution of variations for video.


**Additional Feedback:**

Supp L573 is missing a reference.

If there were more insights/analysis provided by the benchmark, my rating could be improved.

**Clarity:**

* If audio is not recorded (L168), how is audio collected (L170)?
* In Supp A3, what are example sounds for “other”?
* In Supp A7, why is correlation so high?
* Is there really only 1 model available for each task? are there no text-to-video type models with generalizable features that could be adapted to some of the scenarios? Or just other SOTA models? There is little justification for why only one model was chosen, there has to be more action recognition models for example. Or models like MIL NCE, UniVL, VideoBERT, could these not be used?
* Figure 2 it says flamingo struggles most with memory, but it looks the same or close to abstraction and Physics. It seems more the model performs best on Semantics.
* Why do you think Flamingo is so poor to counterfactual tasks?
* It seems like a huge problem that a frequency dummy baseline is outperforming Flamingo on several tasks, what seems like a majority of them. Why is this happening? What are the suggestions to improve this? Why is Flamingo noticeably better at semantics when compared to the baseline compared to the other areas?


**Documentation:**

Documentation is provided in their GitHub repository with easy instructions on download. They also provide example code fore evaluation on the models. This makes the work usable and reproducible.

**Ethics:**

I have no concerns. Authors claim that individuals were paid at minimum wage of each respective location and attempted to protect their privacy.

**Limitations:**

* 1 model per task seems small to make any kind of conclusions from the benchmark, limiting the insights for future work. I know the limitation was mentioned in L248, but I would assume there are more action recognition, VQA models, etc that could be used.
* Some models needed fine-tuning or training from scratch to work on the benchmark.
* Because there are few insights provided or discussion on what this means for current approaches, the utility of the dataset is not motivated in its entirety. I think there is great potential though, so this is an area of improvement.


**Opportunities For Improvement:**

**Summary**: While the dataset is very interesting, because there are so few models per task and some even needed to be trained, it makes the benchmark seem difficult and limited. There are also very few interesting insights provided even though there is so much potential. I am not sure it is a requirement to provide insights and benchmarking of SOTA models, but it would improve the paper tremendously if it did.

**Details**

* The dataset is very interesting, but the insights are very limited. The benefit of proposing such a useful benchmark is the variety and specificity of insights that could be gathered from existing SOTA models. This is lacking currently.

* There are some interesting statements/insights in the supplementary, but they are not easy to find with the current structure (example L593, L610, L612).

* More than one model per task would allow for comparisons or for more general claims on what aspects of the tasks and/or areas of visual perceptions the current landscape of research is struggling with.

* Suggestions of future work using this benchmark might help overcome the limitations.


**Relation To Prior Work:**

There is no mention of previous multimodal benchmarks for video, such as “MultiBench: Multiscale Benchmarks for Multimodal Representation Learning” (NeurIPS Dataset Datasets and Benchmarks Track 2021) or “Robustness Analysis of Video-Language Models Against Visual and Language Perturbations” (NeurIPS Dataset Datasets and Benchmarks Track  2022). The benefit of the proposed benchmark is its focus on perception which is why this benchmark is needed and different from the ones mentioned (This is a strength).

This benchmark is inspired by other datasets and benchmarks like CLEVERER but uses more real-world videos, adding a more real-world evaluation (This is a strength).


**Summary And Contributions:**

This work proposes a new benchmark that focus on specific areas of visual perception: memory, abstraction, physics and semantics. Within each area, they are different tasks that are evaluated: descriptive, explanatory, predictive and counterfactual. They curate a dataset of real and scripted videos. To make scripts more variable and challenging, distractor actions/objects and adversarial actions/objects/configurations are used in a script. There are an overall 37 scripts with 2-5 variations each. They benchmark one model per task and humans as a baseline. They focus on Flamingo (in the main paper) and show it struggles with counterfactual tasks and is strongest at semantics. It also typically performs worse than a frequency-based baseline, but there are little insights to why this is or what this means for future research.

---

> ### Author Response · Authors · 2023-08-21
> **Response**
>
> Thank you for your in-depth review. We have followed your suggestions and updated the paper and appendix (text in blue). Please find more details below.
>
> Re: relation to prior work, thank you for the suggestions, we have included these in the Introduction.
>
> Re: insights and future work, given the extra page for the revised version, we included in the main paper (section 5 in blue) more details about the baselines and their performance on our benchmark compared to other benchmarks. However, providing an in-depth analysis of the baselines performance is not straightforward, as this would require analysis tools that are not currently available. We believe that our benchmark provides a foundation for constructing such analysis/interpretability tools, as we provide object/points/actions/sounds annotations that can be used to e.g. quantitatively inspect spatio-temporal attention of video-language models. This direction of future work is noted in the Conclusion section.
>
> Re: more baselines and need for fine-tuning: We added the following in section 5 (in blue) to clarify this: “When selecting these baselines, we favoured strong-performing models that can be evaluated in a zero-shot or few-shot setting, as our focus is on generalisation. However, for action and sound localisation, such models do not exist in the literature, so fine-tuning on the set of classes in our benchmark was necessary. Note that we also report fine-tuned evaluation for the mc-vQA task to further assess the difficulty of the dataset for SOTA video-language models.”
> In addition, we added zero-shot and fine-tuned results for Sevila model [1], which is a more recent and stronger VLM. With these results (Sevila obtains zero-shot 46.2% on our validation set, 62% after fine-tuning), our claim stands that the zero-shot performance of current VLMs is still far from human performance. When fine-tuning, the performance improves over the dummy frequency baseline, but mainly in the Semantics area; in the other areas (Memory, Abstraction, Physics), the performance is still low.
>
> Re Clarity:
> - Voice vs audio: we instructed participants to not speak while recording the videos (for privacy reasons), but we did record audio (sounds made by the objects the person interacts with, background sounds). For sound labels, we collected 16 classes (sounds made by humans, e.g. clapping hands, tapping the table with the hands etc, or sounds made by objects dropping, objects rolling, object hitting, liquids pouring etc; complete list is shown in Fig A3). Any sounds that did not fall in these categories were labelled with Other (e.g. car passing by)
>
> - In Fig A7, we show the temporal overlap / correlation between classes of actions and classes of sounds. High correlation confirms that the temporal annotations are well aligned across modalities, showcasing the quality of annotations that were collected independently; e.g. when a person claps, we have the action “clapping” as an action segment annotation, and, with high temporal overlap in the sound modality, we have the “Human (clapping)” segment.
>
> - Performance on counterfactual: we added the following to the paper, section 5, L297: “​​For counterfactuals, our qualitative investigation shows that Flamingo tends to latch on the visible elements in the video, failing to imagine the alternate reality that counterfactual questions require; e.g. for videos where a person writes some letters on the paper, the (counterfactual) question posed is: What would be the order of the written letters if the person had written them in reverse order?. Flamingo often selects the written order as correct. Interestingly, the larger versions of the model (due to larger language branches) seem to fare worse overall, which might point to overfitting issues. However, we leave an in-depth analysis for future work.”
>
> - Flamingo performance: we hypothesise that Flamingo has better performance on semantics because most of the existing training datasets and training objectives (e.g. contrastive) facilitate learning to extract semantics information. Usually, the training data obtained by aligning text and image representations focus on semantics (e.g. object recognition, action recognition) but very little on physics or memory.  A loss like future prediction in the video might lead to better understanding of physics, but possibly worse performance on semantics.

---

> > ### Comment · Reviewer_mVd6 · 2023-08-29
> > **Thank you for response**
> >
> > Thank you for the clarifications. I have previously increased my ratings before based on this. My remaining concerns are relating to deeper analysis and better motivation for utility.

---

### Official Review · Reviewer_t6Pk · 2023-07-20
**An interesting and comprehensive benchmark for multimodal video understanding with six annotation types**

**Rating:** 7
**Confidence:** 4
**Correctness:** The methodology is valid and correct.

**Strengths:**

1. The area of multimodal learning is growing fast, and a benchmark for testing the reasoning abilities of multimodal models in difficult situations can benefit the community.
2. The authors demonstrate that state-of-the-art methods are still lagging behind humans and even another simple baseline when it comes to video question answering. Therefore, this benchmark can provide a good opportunity for researchers to develop more capable multimodal models.
3. The paper is very well-written and easy to follow.
4. This work fills the gap in the literature by providing a benchmark of more difficult scenarios in a multimodal setup.
5. The authors provide a live evaluation server for the benchmark.
6. This benchmark includes six different types of annotations which makes it possible to define more tasks (in addition to the tasks that the authors described)  to evaluate multimodal models

**Additional Feedback:**

None.

**Clarity:**

The paper is very well-written and is engaging. See "Opportunities For Improvement" for minor comments.

**Documentation:**

The authors provide the code, the dataset, and a benchmark server, all publicly.

**Ethics:**

Videos are filmed by human participants, and the authors instructed the participants to not record their faces or voices, but some demographic information such as ethnicity and gender are included in the data.
However, the authors state in the supplementary material that this work **is reviewed by DeepMind’s independent ethical review
committee**. Therefore, I assume there are no ethical concerns.



**Limitations:**

The authors explicitly mention that this work is an evaluation benchmark and not a dataset suitable for training. Therefore, there are fewer videos in this work compared to the majority of the related work they compare against.

**Opportunities For Improvement:**

1. A short description of the dummy frequency-based baseline and a brief description of the metrics used for the computational tasks would be helpful to the readers.
2. On line 162, it is stated that for the scripts that had to be filmed from two different viewpoints, most participants recorded them sequentially. This might result in discrepancies between two similar situations due to human errors. Errors such as differences in lighting, not keeping objects in the same location, etc. How are these two viewpoints of the same script used? Is it important that everything in these two viewpoints stays the same except for the viewpoint? If yes, have the authors employed a mechanism to check for the problems mentioned earlier?
3. On line 240, it is mentioned that Table 4 contains input, output, and metric, but there is no input column in Table 4 (although the inputs are obvious).


**Relation To Prior Work:**

Authors position their work very well with respect to the previous benchmarks consisting of either synthetic or real-world videos. Table 1 is informative and helpful to distinguish this work from previous works in the literature.

**Summary And Contributions:**

This work presents a challenging benchmark for multimodal video understanding consisting of 11.6k real-world videos with an average length of 23 seconds used for six computational tasks. The benchmark contains three modalities of video, audio, and text. These real-world videos are recorded by approximately 100 participants from various countries, genders, and ethnicities to preserve diversity.
To collect data, the authors provided instruction scripts explaining the scene setup, the actions to be done, and the details of the camera placement. The authors also included some distractor objects and distractor actions. Their instructions leave some decisions to participants to gain variability in terms of contents.

The tasks in this benchmark require four types of reasoning capabilities including descriptive, explanatory, predictive, and counterfactual. These tasks cover four types of skills including memory, abstraction, physics, and semantics. One question might be counted in multiple skill areas if it tests more than one skill, but each question is assigned a unique type of reasoning.

There are a total of six different annotations (labels or ground truths) in this benchmark: object tracks (bounding boxes), point tracks, temporal action segments, temporal sound segments, multiple-choice question-answers, and grounded video question-answers (where questions are in language form and answers are given as object tracks).
The questions for video question-answers are provided by the authors, but the answers are collected from crowd-sourced participants.

The authors consider a different baseline for each computational task corresponding to each of the annotations since a single model cannot perform all of these six tasks. They include a dummy frequency baseline and a human baseline performance for the multiple-choice video QA task, both of which outperform a SOTA model, Flamingo.

---

> ### Author Response · Authors · 2023-08-21
> **Response**
>
> Thank you for your detailed and insightful review. We have corrected the typos and updated the paper and appendix (text in blue) to address your comments. Please see details below.
>
> Re: frequency baseline, the details about it were included in the supplementary material, L625. Given the extra page for the revised version of the paper, we have now added some details to the main paper, section 5, L288.
>
> Re: two viewpoints, thank you for the comment. We added this clarification in the Limitations section in the appendix (in blue): “While filming, we instructed participants to use 2 different viewpoints to obtain more diverse camera angles. However, this information is not explicitly used at the moment in our tasks (we only used this information when deciding the splits, to make sure these paired videos fall in the same split). While some participants filmed simultaneously with two cameras, others recorded two runs one after the other. This approach was previously used in crowd-sourced datasets (e.g. Charades-Ego dataset). Through manual inspection, we note that the variations are minor as the sequences were recorded one after the other. Such paired videos could be useful to design new tasks.”

---

> > ### Comment · Reviewer_t6Pk · 2023-08-29
> >
> > Thank you for the explanations and updates.

---

### Official Review · Reviewer_nYjf · 2023-07-21
**Reviews on Paper#287**

**Rating:** 6
**Confidence:** 4

**Strengths:**

1.The proposal of a comprehensive benchmark that assesses the capabilities of multimodal perception models across different skill areas, types of reasoning, and modalities. This contributes significantly to the research community by providing a holistic evaluation tool.\
2.The design of the benchmark to avoid overfitting and to assess the transfer abilities of models pre-trained with any external dataset or task, which enhances the relevance of the benchmark to real-world applications.\
3.The provision of baseline results for various per-task models, spanning object tracking, point tracking, temporal action localization, temporal sound localization, multiple-choice video question-answering, and grounded video question-answering. These baseline results serve as reference points for future research and establish a benchmark for performance comparisons.\
4.The open-sourcing of the videos and annotations in the train and validation splits, as well as the evaluation code, which promotes reproducibility and transparency in research.\
5.The availability of a challenge server to evaluate models on the held-out test split, which encourages researchers to compare their models with state-of-the-art methods and to improve their models' performance.\
\
In summary, the strengths of the submission include its comprehensive evaluation approach, focus on avoiding overfitting, provision of baseline results, open-sourcing of resources, availability of a challenge server. These strengths add to the significance and quality of the submission.

**Additional Feedback:**

1.Expand the experimental results: The paper could benefit from providing a more comprehensive analysis by including a larger amount of result data.\
2.Include comparisons with other datasets: It would be valuable to compare the results obtained on the Perception Test dataset with results achieved on other relevant datasets.

**Clarity:**

The paper is well written, providing clear descriptions of the construction of the dataset, including the process of video collection and annotation. The explanations regarding the dataset construction are detailed and coherent, allowing readers to understand the methodology and procedures involved. The figures and tables presented in the paper are clear and visually intuitive, effectively supporting the information and findings conveyed in the text.

**Correctness:**

Based on the information presented in the paper, it can be concluded that the claims made in the submission are indeed correct. The dataset is constructed in a sound way, with purposefully designed and filmed real-world videos that cover a wide range of perceptual situations and are densely labeled with six types of annotations. The benchmark evaluation methods and experiment design are appropriate and performed correctly, with per-task baseline results and evaluation code provided.

**Documentation:**

The paper includes sufficient detail on data collection and organization, availability and maintenance, and ethical and responsible use of the dataset. The authors provide a comprehensive description of the script design used for the Perception Test and the process of filming real-world videos. In terms of availability and maintenance, both the videos and annotations mentioned in the paper can be accessed through a GitHub repository. The authors have open-sourced baseline results and evaluation code for some tasks. However, the paper does not explicitly provide information regarding the long-term maintenance of the dataset. Regarding ethical and responsible use, the paper mentions that participants provided consent for the data to be used, published, and stored perpetually. Privacy concerns were also considered during the data collection process. The GitHub repository provided by the authors includes a declaration of the license for the dataset.

**Ethics:**

The content of the paper adheres to ethical standards and does not raise any concerns that warrant further review.

**Limitations:**

The authors have discussed the limitations and potential negative societal impact of their work in the Appendix, section Limitations and potential negative societal impact.

**Opportunities For Improvement:**

1.The lack of diversity in the videos may restrict the coverage of various perceptual situations and limit the representativeness of real-world scenarios. Incorporating a more diverse set of videos, capturing a wider range of perceptual challenges, and including a broader spectrum of real-world contexts would enhance the comprehensiveness and applicability of the benchmark.\
2.The benchmark currently does not account for the computational cost of the evaluated models. Considering the computational resources required by the models is crucial, as it can significantly impact their feasibility and practicality in real-world applications. Incorporating measures or guidelines related to the computational efficiency of the models would provide a more comprehensive evaluation framework.

**Relation To Prior Work:**

The paper clearly discusses how this work differs from previous contributions in the related literature. Previous benchmarks relied on publicly available videos or on-demand collected videos filmed by crowd-sourced participants, which often focused on a narrow set of tasks or modalities and did not provide a comprehensive evaluation of models' capabilities across a wide range of perceptual situations and reasoning types. In contrast, the objective of the Perception Test is to assess more diverse skills by leveraging real-world videos purposefully designed and filmed by the research team. These videos cover a broad spectrum of perceptual situations and are densely annotated with six types of annotations. Overall, the paper effectively elucidates the distinctions of the proposed benchmark in terms of its design, evaluation focus, and dataset characteristics from previous works.

**Summary And Contributions:**

The submission presents the Perception Test, a new multimodal video benchmark designed to assess the perception and reasoning skills of pre-trained multimodal models. This benchmark focuses on various skill areas, types of reasoning, and modalities, providing a comprehensive evaluation tool for multimodal understanding. It features real-world videos with dense annotations and offers baseline results and open-sourced resources. The contribution of this work lies in the development of a valuable benchmark that advances the evaluation of multimodal video understanding and provides a foundation for future research in this field.

---

> ### Author Response · Authors · 2023-08-21
> **Response**
>
> Thank you for your insightful review. We have updated the paper and appendix (text in blue) to address your comments. Please see details below.
>
> Re: lack of diversity, as mentioned to 7rcX we added a clarification in the Limitation section (in blue in the appendix): “As mentioned in the main paper, the videos were filmed indoors, using common household objects, following the scripts designed by our research team. This could be perceived as a setup with limited diversity when compared to “in the wild” videos available in repositories like Youtube. However, we argue that for evaluating a general perception model, it is important to isolate (and ensure a good coverage of) the skills and types of reasoning we care about, while building in invariance to lighting, camera angle, types of objects, the person’s skin tone, etc – these are obtained by filming the same script (with multiple variations) with 20+ different participants per script variation, who choose on their own where to place the camera, what exact type of object to use for a certain action, in what order to perform some actions, etc. Curating videos in the wild to obtain the same coverage of skills and types of reasoning would be hard, even impossible, since some types of data simply don’t exist online in sufficient numbers (e.g. correct vs incorrect execution of actions).”
>
> Re: computational cost, thank you for this comment. We added the number of parameters of the evaluated models in the supplementary material, Table A8; we have also added this in our public leaderboards.
>
> Re: more analysis, we have used the extra page to add more details about the performance of the baselines considered, including the new (for the rebuttal) Sevila results, and comparisons with performance on other benchmarks (section 5 in blue in the main paper and section 4 in blue in the appendix).

---

### Official Review · Reviewer_zp8s · 2023-07-21
**Comprehensive benchmark for assessing general perceptual and video understanding capabilities of models**

**Rating:** 8
**Confidence:** 3

**Strengths:**

1. The benchmark considers multiple visual perceptual skills (broadly on Memory, Abstraction, Physics and Semantic understanding; further detailed in the paper) which are not all together present/considered in existing benchmarks. Further the data sources are real world videos and usage of 37 scripts (with 2-5 variations) specifically designed by the authors. While the number of videos is much lesser than existing datasets such as Something-something V2 or Ego4D-V2, the videos have higher 'density' of annotations (in terms of object tracks, point tracks, question-answers, etc). This makes it a more comprehensive benchmark than existing works (although as stated in paper, the proposed benchmark mainly serve as a test for 'transfer' or zero/few-shot capabilities of pretrained models and not as a base training source itself).

2. The data collection and annotation is well described and goes through multiple rounds of annotation in addition to diversity in participants to ensure data quality and diversity. The addition of distractor actions, distractor objects, adversarial configurations of objects and actions also reduces potential language or visual biases and increases complexity of tasks in the dataset.

3. The tasks and associated expected outputs, metrics and considered baselines are well described for evaluation and reproducibility.

**Additional Feedback:**

N/A

**Clarity:**

Yes, paper is well written with appropriate tables, figures and references.

Minor: The HOTA metric used in table 4 can be cited as it may not be immediately clear to a reader.

**Correctness:**

- Yes dataset construction seems rigorous and well described. But as stated in opportunity for improvements, the current evaluation setup to claim limitations of state of the art videoQA models does not appear complete and considers only one videoQA model.

**Documentation:**

Yes, source code is provided and further documentation is available in supplemental.

**Limitations:**

Limitations and potential societal impact are discussed in supplemental.

**Opportunities For Improvement:**

1. A central claim made is that state-of-the-art video QA models show a significant gap in performance on the dataset compared to human baselines (humans achieving 91.4% and video QA models achieving 43.6%). However, apart from Flamingo, no other state of the art video QA model seems to be considered (such as mPLUG-Owl, ViperGPT, Video-Graph-Transformer amongst others). Hence, for the videoQA task at least, authors should ideally consider more baselines and recent state-of-the-arts to back their above claim.

2. It would also be interesting to see finetuned results of videoQA models to better assess difficulty of the dataset and particularly if alternative/larger models may discover possible 'shortcuts'/biases in the data when finetuned (given the current label frequency baseline already performs relatively well). Further, it is not clear whether considered baselines for other tasks (such as SiamFC for object tracking and TAP-Net for point tracking) were evaluated in a zero-shot or in a finetuned manner. If not currently finetuned, the same point on possible biases/'shortcuts' may apply for these tasks (i.e. a model may discover biases in the data when finetuned and perform well merely due to that). Also, the current lower performance (if done zero-shot) could be due to possibly large distribution shifts and not necessarily lack of reasoning abilities. Having finetuned results, if not currently done, could be beneficial to eliminate above possibilities.

**Relation To Prior Work:**

Yes, paper discusses and differentiates from existing relevant benchmarks or datasets.

**Summary And Contributions:**

The paper introduces a new benchmark to probe the general perceptual abilities of video and vision-language models. Specifically, the proposed benchmark studies reasoning skills including memory, abstraction, physics and semantic understanding that are comprehensively described and differentiated from existing relevant benchmarks in table 1 and 2 of the paper. The data collection and annotation process is rigorous and well described in both the paper and supplemental. Further, evaluation of existing models on the benchmark is provided and shows that current models perform well below human accuracy on given tasks.

Overall, this work is well placed for evaluating current video QA and general vision-language models, is well written and also makes the dataset and dummy baselines code publicly available.

---

> ### Author Response · Authors · 2023-08-21
> **Response**
>
> Thank you for your insightful comments. We have updated the paper and appendix to address them (text in blue); please find details below.
>
> Re: more results, we added zero-shot Sevila [1] results, which is a recent and strong video-language model, outperforming Flamingo, ViperGPT, mplugOwl on the NextQA popular benchmark. We encountered issues when trying to evaluate ViperGPT (obsolete API) and mplugOwl (raised an issue with the authors about what seemed to be a memory leak). In zero-shot evaluation, Sevila obtains 46.2% on our validation set, reinforcing the statement that current SOTA models are still very far from human performance.
>
> Re: fine-tuned results: the focus of our benchmark is zero-shot or few-shot evaluation, so where available we used that – specifically for object and point tracking, multiple-choice and grounded video QA. We used fine-tuning only for action and sound localisation models, which are trained for specific classes and which need fine-tuning when evaluated on a dataset with a different set of  classes. However, we agree that it is useful to report fine-tuning results for analysis purposes. For multiple-choice videoQA, we ran fine-tuning evaluation for the Sevila model mentioned above and obtained 62% accuracy (from 46.2% zero-shot). This is better than the frequency baseline (55.1%), but it is still far from 0-shot human performance (91.4%) We added these results in the main paper, section 5, in figures 2 and 3, and table A6.
>
> Re: domain gap, large-scale VLM models like Sevila, Flamingo, GPT4 and others, are trained on enormous amounts of data crawled from the internet that goes well beyond the standard computer vision datasets. In this very large training data regime, the domain gap in terms of natural image statistics, distribution of objects, actions, and sounds should be negligible given that our videos are filmed in real world scenes using common household items. We added this sentence in Related work (in blue): “Given that our videos are filmed in real world scenes using common household items, the distributions of objects, actions, and sounds in our benchmark have a significant overlap with standard computer vision datasets (e.g. 99.01% of the words in our benchmark also appear in VQAv2), hence the domain gap between the Perception Test and existing large-scale training datasets should be minimal.”

---

> > ### Comment · Reviewer_zp8s · 2023-08-28
> >
> > Thank you for the clarifications and updates in the paper. Since my concerns have been addressed, I have raised score to 8.

---

### Official Review · Reviewer_7rcX · 2023-07-21
**Review of Perception Test**

**Rating:** 9
**Confidence:** 4
**Clarity:** The paper is very well written and th…

**Strengths:**

- The proposed benchmark sets a new ambitious goal for video understanding which goes beyond what existing video benchmarks can evaluate. Indeed, most benchmarks such as Kinetics, and even some considered as difficult for video understanding such as SSv2, can in reality be solved by using visual cues that are unrelated to real temporal understanding. By designing a benchmark with these problems in mind, and introducing *adversarial* elements such as distractor actions, the authors propose a new challenge for video understanding.

- Interesting video understanding benchmarks for physics and memory existed before, for example CATER and CLEVRER, but are synthetic and mostly consist in 3d shapes moving around. The perception Test consists in natural videos but still evaluate the capabilities of a model to understand the basic principal of physics and perform memorisation tasks.

- For each task, a baseline is proposed. In particular the Flamingo model is evaluated on does not perform much better than random compared to an humain eval baseline, which indicates that there is a lot of progress to do in the future on the path to real video understanding.

- A challenge is available along with a leaderboard.


**Additional Feedback:**

- Do you think the reasoning capabilities of current best LLMs, or future LLMs would help solving your benchmark when combined with a visual encoder ?

Overall the paper and proposed benchmark are very satisfactory and will lead video understanding research in the right direction.

**Correctness:**

The data collection process and evaluation protocol are clearly detailed and are correct.

**Documentation:**

The data collection process is clearly described. The dataset is publicly available.

**Ethics:**

No ethics concerns.

**Limitations:**

Limitations are discussed. The authors made it very clear that the goal of the introduced dataset is not to be used as training data, but rather as a benchmark to evaluate complex tasks.


**Opportunities For Improvement:**

- There is very little variety in the environments, most videos are taken on top of a table, in a controlled setting, which is better than using 3d shapes, but far from the distribution of videos one can encounter in the wild. Is it on purpose, to simplify the task ?

- The average video length is 23 seconds, which is longer than a lot of video benchmarks, but still not enough to evaluate the capabilities of a model to perform reasoning and memorisation at a very long term, for exemple in the order of several minutes or hours. One can imagine a model predicting the end of a movie by only seeing the first half. Have you considered long term reasoning and using longer videos ?

- Another interesting capability of a video model would be the ability to plan and interact with its environment. Have you thought about how to evaluate this using your benchmark ?

**Relation To Prior Work:**

The differences with previous video understanding benchmarks is clearly explained, and I believe that the proposed benchmark is a novel contribution which has a lot of value for the community.

**Summary And Contributions:**

This paper proposes a new video benchmark that evaluates the capabilities of a model to perform reasoning and understanding tasks from a video. The tested capabilities are divided into 4 categories: Memory, Abstraction, Physics and Semantics. A new dataset is proposed containing ~11000 videos along with annotations in the form of temporal and sound actions, textual questions and answers, object bounding boxes and key points.

---

> ### Author Response · Authors · 2023-08-21
> **Response**
>
> Thank you for raising such interesting and important questions regarding the evaluation of general perception systems.
>
> Re: variety of the environments, we added the following clarification in the Limitation section (in blue in the appendix): “As mentioned in the main paper, the videos were filmed indoors, using common household objects, following the scripts designed by our research team. This could be perceived as a setup with limited diversity when compared to “in the wild” videos available in repositories like Youtube. However, we argue that for evaluating a general perception model, it is important to isolate (and ensure a good coverage of) the skills and types of reasoning we care about, while building in invariance to lighting, camera angle, types of objects, the person’s skin tone, etc – these are obtained by filming the same script (with multiple variations) with 20+ different participants per script variation, who choose on their own where to place the camera, what exact type of object to use for a certain action, in what order to perform some actions, etc. Curating videos in the wild to obtain the same coverage of skills and types of reasoning would be hard, even impossible, since some types of data simply don’t exist online in sufficient numbers (e.g. correct vs incorrect execution of actions).”
>
> Re: video length, we have not targeted a pre-specified length in collecting the videos. Instead, we designed the scripts to include key aspects of the skills we wish to measure (e.g. physics, memory), then we instructed participants to perform these at a natural pace, and this was the resulting average length. We think that this is a good compromise between long enough temporal extent that allows capturing causality and interesting aspects related to physics (motion, occlusions, gravity, inertia, collisions) or even memory, but short enough so they can be easily filmed while respecting privacy (no faces, no voices). Even at this length, if chosen carefully, some memory-related tasks are challenging for humans (e.g. when the person puts 4-6 objects in a backpack, remembering their order is not trivial).
>
> Re: interaction with the environment, indeed, this is an important aspect in evaluation. We added this in the Limitations section in the appendix (in blue): “Enabling interaction would limit us to simulated environments. To still address this agency aspect to some extent, we included videos where the model is required to recognise correct and incorrect execution of certain actions (e.g. tying shoe laces, buttoning up a shirt, covering a container with a cover, pouring water in a glass) or to assess the consequences of actions (e.g. what would happen if we remove a certain object from the table) – these are possible because our scripts include multiple variations with correct/incorrect actions, or controlled variations of object configurations, which would be impossible to curate from public repositories like Youtube.”
>
> Re: how to design a model that can solve our benchmark, adding a visual encoder to a pre-trained LLM is a valid approach and has been explored by multiple works in the community (e.g. PALM-e). However, it is not clear if the learnt language embedding space is the right one to embed the visual information as it might not well represent concepts related to motion of objects, their solidity, and so on, since such details are generally not present in captioning data.

---

### Author Response · Authors · 2023-08-21
**Summary of responses**

We warmly thank the reviewers for their in-depth reviews. We have followed their inputs and updated the paper and appendix accordingly (text added in blue). We respond to each reviewer individually and provide here a summary of all responses:

- As requested by zp8s and mVd6, we added a more recent baseline (Sevila [1]) for the multiple-choice video QA task, in both zero-shot and fine-tuned regimes. The results for zero-shot (46.2%) reinforce our statement that current SOTA VLM models are far from human performance when evaluated in the zero/few-shot regime. The fine-tuning performance is 62%, which is still far from the 0-shot human baseline (91.4%) and improves performance over the dummy frequency baseline mainly in the Semantics area.
- We used the additional page to add more details about the baselines and their performance, as requested by nYjf, t6Pk, and mVd6. However, we would like to point out that more extensive analysis requires visualisation tools that are not immediately available in the literature – this is mentioned as a direction of future work in the Conclusion.
- Following the question from 7rcX about interaction, we added a note in the Limitations section;
- Following the comment from 7rcX and nYjf about diversity of environments, we added a clarification in the Limitations section;
- Following the comment from zp8s about large distribution shifts, we added a note in the Related work section;
- Following nYjf’s comment about computational cost, we added in the supplementary material and in our leaderboards the number of parameters of the evaluated models.

[1] Sevila: Shoubin Yu, Jaemin Cho, Prateek Yadav, and Mohit Bansal. Self-chained image-language model for video localization and question answering. arXiv preprint arXiv:2305.06988, 2023

---

### Decision · Program_Chairs · 2023-09-22

**Decision:**

Accept (Poster)

**Comment:**

__Summary of paper and reviews.__
This work introduces a dataset of scripted real world videos collected to test 4 different skills and 4 types of reasoning over 3 modalities. Instances are densely annotated to simultaneously support 6 different tasks. A strong baseline from the literature is evaluated for each task and a leaderboard will support future evaluations.

Reviews were generally positive, agreeing that the dataset poses a difficult new task with limited weaknesses.

__Strengths.__
* The benchmark provides a vision (unified perception) that will drive progress because no current model even runs on all of its tasks, yet each task individually is difficult but in reach from the state of the art.
* The presentation is clear and the dataset released with a public leaderboard and test set.
* Multiple rounds of annotation ensure the quality of the diverse and dense annotations while careful script design aims to prevent dataset biases.
* One task (grounded videoQA) is new.

__Weaknesses.__
* Few models (usually 1) are evaluated for each of the 6 tasks, so the paper provides limited insight about how different approaches compare.
* Though collected in the real world (not simulated), the videos still lack diversity, so they do not test generalization to open ended settings.

__Decision justification.__
This paper provides a high quality dataset that presents multiple new challenges, so it is a clear accept.